# Non-cell-autonomous control of mouse gastruloid development by the ultra-conserved lncRNA *T-UCstem1*

Arianna Coppola[1,2], Filomena Amoroso[1,2], Federica Saracino [1], Gennaro Andolfi[1], Edoardo Sozzi [3], Paolo Salerno[4], Pietro Zoppoli[4], Alessandro Fiorenzano [1,3,4], Giuseppe Merla [4,5], Eduardo Jorge Patriarca[1], Gabriella Minchiotti [1✉] & Annalisa Fico [1✉]

## Abstract

The role of long non-coding RNAs (lncRNAs) in early mammalian embryogenesis remains unclear due to the complexity of the regulatory mechanisms involving lncRNAs and the limited availability of embryo samples. Stem cell-based models, such as mouse gastruloids, provide new ways to address these challenges. Here, we investigate the function of ultra-conserved lncRNA *T-UCstem1* in mammalian body plan formation. Combining morphological and immunofluorescence imaging analyses with bulk and single-cell transcriptomics, we provide evidence indicating that *T-UCstem1* is important for mouse gastruloid development and anteroposterior axis extension. *T-UCstem1* depletion in gastruloids results in their aberrant development, with defects in the expression of differentiation markers and persistence of pluripotency gene expression. Our single-cell analyses reveal higher levels of cellular heterogeneity in *T-UCstem1*-knockdown gastruloids. The presence of cell populations co-expressing pluripotency and differentiation markers points to an important role of *T-UCstem1* in the establishment and maintenance of cell identity. Mechanistically, we show that *T-UCstem1* acts in a non-cell-autonomous manner through regulation of the Dickkopf-related protein 1 (DKK1)-dependent WNT pathway. Our study identifies a new role for an ultra-conserved lncRNA in gastruloid development and highlights gastruloids as a model system for studying lncRNAs in early development.

**Keywords** Noncoding RNAs; *T-UCstem1*; Gastruloids; DKK1; WNT Pathway
**Subject Categories** Development; Signal Transduction; Stem Cells & Regenerative Medicine

## Introduction

The transformation of a symmetric embryo into a structured body plan, which serves as the foundation for an adult organism, hinges on the essential process of breaking embryonic symmetry. This pivotal event is guided by morphogen signalling gradients that direct the formation of the anteroposterior axis. However, the exact triggers and the spatiotemporal establishment of these signalling gradients in mammalian embryos remain largely elusive. Stem cell-based in vitro models of embryogenesis provide a unique opportunity to quantitatively analyze the diverse physical, cellular and molecular mechanisms involved in shaping the mammalian embryo (Harrison et al, 2017; Martinez Arias et al, 2024; Rivron et al, 2018; Sozen et al, 2018; Sozen et al, 2019; Veenvliet et al, 2020; Zhang et al, 2019). Among the stem cell-based embryo models described so far, gastruloids are emerging as a robust model system for dissecting the cellular and molecular mechanisms that underline the formation of the mammalian body plan (Beccari et al, 2018; Hashmi et al, 2022; Moris et al, 2020; Turner et al, 2017; van den Brink et al, 2020; van den Brink et al, 2014). However, despite their versatility, gastruloids lack extra-embryonic tissues and do not fully recapitulate anterior brain development, limiting their capacity to model the entire embryonic process (Moris et al, 2020; van den Brink et al, 2014).

Over the past decade, numerous studies have highlighted various aspects of the fine regulation governing the mechanisms underlying gastruloid development. Different studies have emphasized the use of gastruloids to reproduce the effects of gene ablation, such as for Nodal (Turner et al, 2017), Cripto (Cermola et al, 2021) and Tbx6 (Veenvliet et al, 2020), to modulate signalling pathways, such as WNT (Girgin et al, 2021; Turner et al, 2017; van den Brink et al, 2014) and BMP (van den Brink et al, 2014) and for drug safety testing (Amoroso et al, 2023; Cermola et al, 2022; Mantziou et al, 2021), while others have highlighted the importance of mechanical forces in controlling tissue flows (Hashmi et al, 2022). In this context, the role of both short and long non-coding RNAs

[1]Stem Cell Fate Laboratory, Institute of Genetics and Biophysics "A. Buzzati-Traverso", CNR, Naples, Italy. [2]Department of Precision Medicine, University of Campania Luigi Vanvitelli, Naples, Italy. [3]Developmental and Regenerative Neurobiology, Lund Stem Cell Center, Department of Experimental Medical Science, Faculty of Medicine, Lund University, Lund, Sweden. [4]Department of Molecular Medicine and Medical Biotechnology, University of Naples Federico II, Naples, Italy. [5]Laboratory of Regulatory and Functional Genomics, Fondazione IRCCS Casa Sollievo della Sofferenza, San Giovanni Rotondo, Foggia, Italy. ✉E-mail: gabriella.minchiotti@igb.cnr.it; annalisa.fico@igb.cnr.it

  

(ncRNAs) have been mostly unexplored, despite their well-established activity as key regulators of stem cell behavior and embryo development (Fico et al, 2019; Mirzadeh Azad et al, 2021; Pauli et al, 2011; Ulitsky and Bartel, 2013). It is important to note that the specific functions of individual ncRNAs are often difficult to discern in vivo, likely due to compensatory mechanisms and/or maternal contributions (Han et al, 2018; Zhang et al, 2012). Stem cell-based embryo models, like gastruloids, overcome some of these limitations and could represent a powerful tool to tackle these challenges. In this study, we used mouse gastruloids to explore the functional role of a family of long non-coding (lncRNAs), the *transcribed ultra conserved elements* (*T-UCEs*), in the formation of the anteroposterior axis in mammals. This family of molecules has intrigued researchers for over a decade now since, unlike most lncRNAs that exhibit poor conservation, they are transcribed from DNA sequences that are 100% conserved across human, mouse, and rat genomes (Bejerano et al, 2004). Their roles have become increasingly evident especially in pathological conditions such as cancer and various human diseases (Calin et al, 2007; Galasso et al, 2014; Habic et al, 2019; Pereira Zambalde et al, 2020), and, more recently, they are emerging as key players in physiological processes like stemness and cellular differentiation (Fiorenzano et al, 2018; Galasso et al, 2014; Panatta et al, 2020; Pascale et al, 2020; Zhou et al, 2017). However, to our knowledge, the role of *T-UCEs* in early phases of embryonic development has not been explored so far.

Here, we investigated the role of *T-UCstem1*, belonging to the *T-UCE* family (Fiorenzano et al, 2018), and provide unprecedented evidence that its activity is crucial for the proper anteroposterior axis formation in gastruloid development.

# Results

## *T-UCstem1* is essential for proper gastruloid development

*T-UCstem1* is a long non-coding RNA (lncRNA) that harbors the ultraconserved element *uc.170* and exhibits a remarkably high level of sequence conservation across vertebrates along its entire length (Fig. EV1A). *T-UCstem1* has been shown to play a critical role in maintaining self-renewal of embryonic stem cell (ESC) cultured in FBS/LIF (Fiorenzano et al, 2018).

To investigate the potential role of *T-UCstem1* in gastruloid development, we used *T-UCstem1* knockdown (KD) mESCs. First, we evaluated the effect of *T-UCstem1* downregulation on gastruloid formation efficiency (GFE) using two independent *T-UCstem1* KD ESC clones (referred to as KD-1 and KD-2) that have been previously described (Fiorenzano et al, 2018). KD-1, KD-2 and Control (NT, non-targeting) cells were plated at low density (250 cells/cm²) on gelatin-coated plates in FBS/LIF/2i medium, a condition that supports self-renewal of *T-UCstem1* KD ESCs (Fiorenzano et al, 2018). After 5 days in culture, the resulting colonies were dissociated with accutase, and the cells were seeded in 96-well ultra-low attachment plates (150 cells/well) (Fig. 1A). At 48 h after aggregation (AA), both Control and *T-UCstem1* KD cells generated spherical, highly compacted aggregates with a diameter of ~ 170 μm (Fig. 1B,C). Immunofluorescence analysis showed that both Control and KD cell aggregates (48 h AA) stained positive for the pluripotency marker Oct4 and did not express Brachyury

(Fig. 1D). At day 4 after aggregation (96 h AA), a protrusion zone became evident in the majority (~90%) of Control aggregates. In contrast, *T-UCstem1* KD aggregates either retained their initial spherical shape or adopted an irregular shape (Fig. 1B). Later on, at day 5 (120 h AA), while Control aggregates developed into fully elongated gastruloids (>90%, Figs. 1B,E and EV1B), *T-UCstem1* KD aggregates maintained a spheroidal/ovoidal shape and increased in size (Figs. 1B,E and EV1B). Specifically, the length of KD-1 and KD-2 aggregates ranged from 309.7 ± 28.6 μm to 257.5 ± 49.8 μm, and the width from 223.1 ± 22.5 μm to 169.3 ± 27.4 μm, respectively, in contrast to the Control gastruloids, which showed a length of 480.6 ± 61.2 μm and a width of 177.8 ± 70.8 μm. Accordingly, the elongation index was significantly lower in *T-UCstem1* KD gastruloids compared to Control (2.9 ± 0.5 for NT *vs* 1.4 ± 0.2 for KD-1 and 1.6 ± 0.4 for KD-2) (Fig. 1E).

Notably, *T-UCstem1* KD aggregates exhibited remarkable morphological variability, ranging from spherical to ovoidal shapes and from small to large cellular structures that failed to properly elongate into classical gastruloids (Fig. EV1B).

To investigate the impact of *T-UCstem1* on gastruloid development at a molecular level, we examined the expression profiles of pluripotency and differentiation markers in 5-day-old Control and *T-UCstem1* KD gastruloids (Fig. 1F). As expected, immunofluorescence analysis showed few Oct4, Nanog, and Sox2 positive cells localized at the apical zones in Control gastruloids. In contrast, these cells were enriched in KD gastruloids and distributed throughout the aggregates (Fig. 1F). Remarkably, the differentiation marker Brachyury Cdx2, and Sox17, were expressed in both Control and *T-UCstem1* KD gastruloids (Fig. 1F), suggesting that symmetry braking occurs. We next analysed E-Cadherin expression. As expected, E-Cadherin was localised at one pole of Control gastruloids, conversely its expression was mostly diffuse throughout *T-UCstem1* KD gastruloids (Fig. 1F).

Together, these data suggest that *T-UCstem1* is dispensable for the early phases of cell aggregation and compaction, while it is required for gastruloid elongation and exit from pluripotency.

## Increased activation of WNT pathway overcomes the effect of *T-UCstem1* downregulation

It is well known that the efficiency of gastruloid formation is influenced by various factors, including the pluripotency state, the cell density, and the concentration of the WNT pathway inducer CHIR99021 (CHIR) (Amoroso et al, 2023; Beccari et al, 2018; Cermola et al, 2022; Cermola et al, 2021; Turner et al, 2017; Veenvliet et al, 2020). We first compared the pluripotency state of Control and *T-UCstem1* KD cells in a colony formation assay. To this end, both Control and *T-UCstem1* KD ESCs were seeded at low density on gelatin-coated plates, cultured for 5 days in FBS/LIF/2i medium (Fig. 2A) and the resulting colonies were stained with crystal violet. Both Control and KD ESC colonies exhibited a round domed shape morphology characteristic of undifferentiated *naive* ESCs (Fig. 2B). Moreover, they stained positive for the pluripotency markers Oct4 and Nanog, while expression of the differentiation markers Brachyury, Cdx2, Nestin, and Sox17 was not detected as shown by immunofluorescence analysis (Fig. 2C). These findings demonstrate that Control and *T-UCstem1* KD cells in the undifferentiated state show the same level of pluripotency at both the morphological and molecular levels.

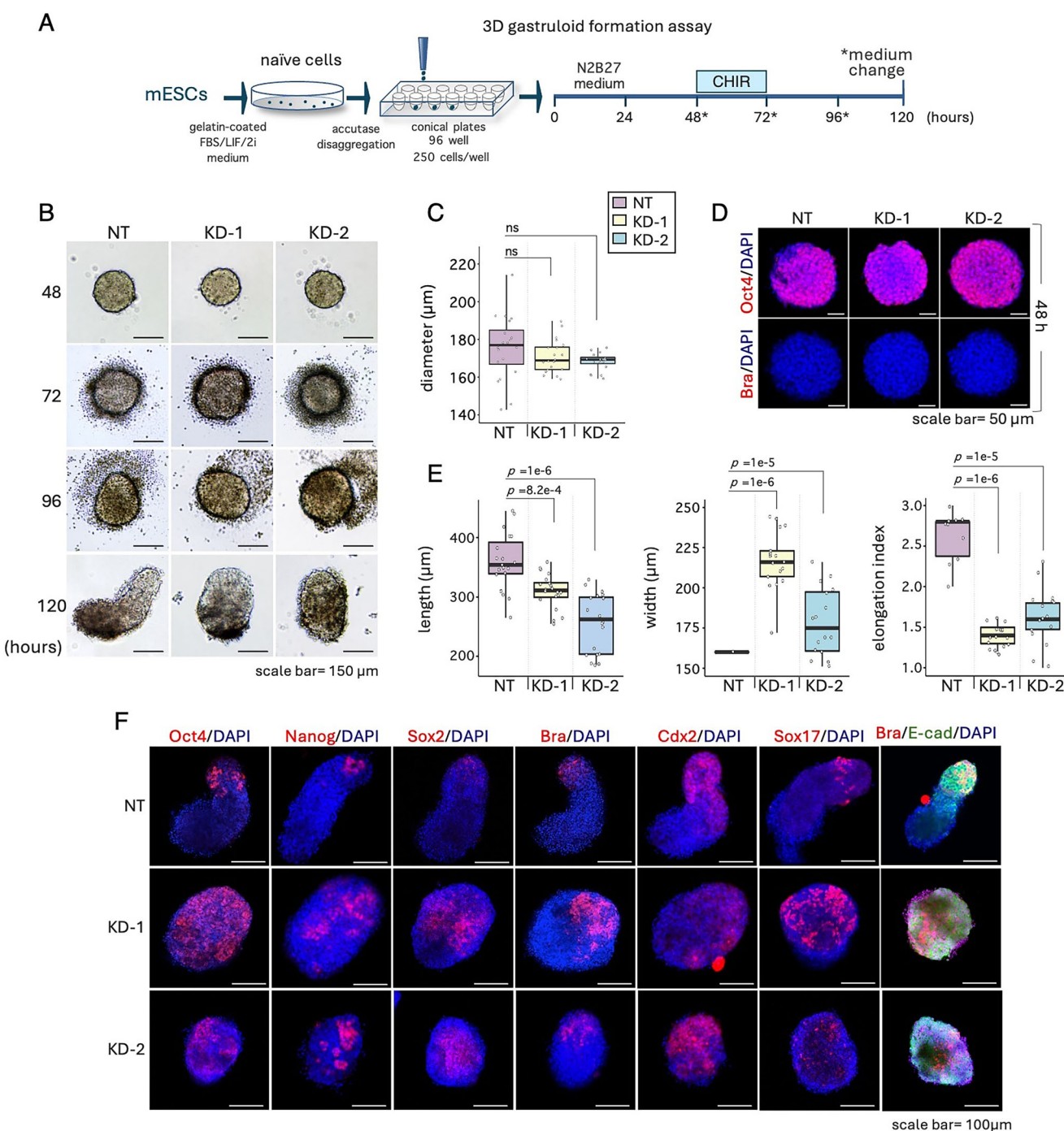

**Figure 1.   *T-UCstem1* is essential for proper gastruloid development.**

(**A**) Schematic representation of the experimental design. Control (NT) and *T-UCstem1* KD mESCs were plated on 96-well ultra-low conical plates. CHIR99021 (3µM) was added between 48 and 72 h; cell aggregates were cultured until 120 h. (**B**) Representative brightfield images of aggregate to gastruloid transition at the indicated time points after aggregation. Scale bar, 150 µm. (**C**) Boxplot diagrams of aggregate diameters at 48 h (n = 3 independent experiments; 20 gastruloids/condition). Data are shown as mean ± SD. Boxplots display the minimum, first quartile, median, third quartile, and maximum. Statistical significance was assessed by one-way ANOVA with Tukey's multiple comparison test. P values of ≤0.05 were considered statistically significant. (**D**) Representative confocal images of gastruloids stained with Oct4 (pluripotency marker) and Brachyury (differentiation marker) at 48 h. Nuclei were counterstained with DAPI (blue). Scale bar, 50 µm. All images are representative of three individual gastruloids. (**E**) Boxplot diagrams of gastruloids length (left), width (middle) and elongation index (right) at 120 h (n = 3 independent experiments; 20 gastruloids/ condition). Data are shown as mean ± SD. Boxplots display the minimum, first quartile, median, third quartile, and maximum. Statistical significance was assessed by one-way ANOVA with Tukey's multiple comparison test. P values of ≤0.05 were considered statistically significant. (**F**) Immunofluorescence analysis of pluripotency (Oct4, Nanog, Sox2), differentiation (Brachyury, Cdx2, Sox17) and cell adhesion (E-Cad) markers expression. Representative confocal images of gastruloids at 120 h. Nuclei were counterstained with DAPI (blue). Scale bar, 100 µm. All images are representative of five individual gastruloids. Source data are available online for this figure.

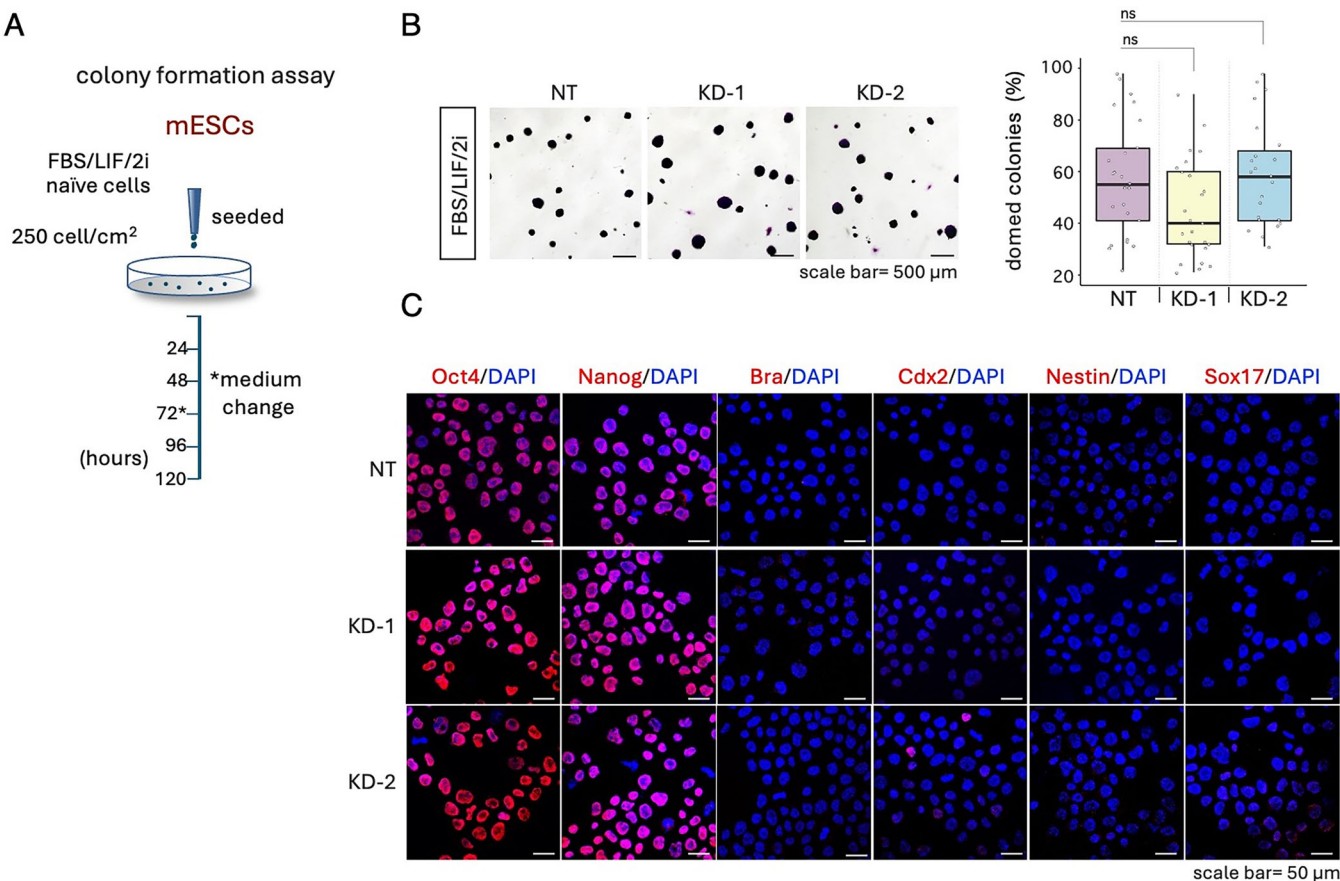

**Figure 2. Pluripotency state of control and *T-UCstem1* KD cells at initial time point.**

(A) Schematic representation of the experimental design. Control (NT) and *T-UCstem1* KD mESCs were plated at low density (250 cells/cm²) in FBS/LIF/2i medium for 5 days. (B) Representative brightfield images (left) of crystal violet stained colonies generated, after 5 days in FBS/LIF/2i and quantification of the fraction (%) of domed colonies (right). Scale bar, 500 μm (n = 5 independent experiments; five fields/condition). Data are shown as mean ± SD. Statistical significance was assessed by one-way ANOVA with Tukey's multiple comparison test. P values of ≤0.05 were considered statistically significant. (C) Representative confocal images of Oct4, Nanog, Brachyury, Cdx2, Nestin and Sox17 staining on cytospin NT, *T-UCstem1* KD-1, *T-UCstem1* KD-2 ESCs after 5 days in FBS/LIF/2i. Nuclei were counterstained with DAPI (blue). Scale bar, 50 μm. All images are representative of three fields/condition. Source data are available online for this figure.

We then evaluated the effect of changing cell density on *T-UCstem1* KD gastruloid formation by increasing the number of ESCs seeded from 100 to 250 cells per well. The diameter of both Control and *T-UCstem1* KD aggregates (48 h AA) increased with the number of seeded cells (Figs. 3A and EV2A), as well as the length and the width of 5-day-old gastruloids (Fig. EV2B,C). However, unlike Controls, the *T-UCstem1* KD aggregates maintained a spheroidal/ovoidal morphology and did not undergo proper elongation (Figs. 3B and EV2D). Consistent with the morphological analysis, immunofluorescence data showed defective expression of pluripotency and differentiation markers in *T-UCstem1* KD gastruloids at all cell density tested (Fig. 3C). These findings support the idea that *T-UCstem1* is dispensable for the initial phase of cell aggregation, but it is crucial for promoting gastruloid elongation, independently of the cell number.

Finally, to rule out the possibility that the concentration of CHIR used (3 μM) might not be appropriate to induce proper elongation of the *T-UCstem1* KD aggregates, we examined the dose-dependent effect of CHIR (3–8 μM) on *T-UCstem1* KD gastruloids (Figs. 4A and EV3A–C). Increased concentrations of

CHIR were able to rescue the ability of both *T-UCstem1* KD-1 and KD-2 ESCs to develop elongated gastruloids. Specifically, at the highest concentration of CHIR, 5 and 6 μM respectively, KD-1 and KD-2 gastruloids elongated (Fig. 4A) and showed proper expression profile of Oct4, Nanog, Cdx2 and Brachyury (Fig. 4B). At the highest concentration (8 μM), both Control and *T-UCstem1* KD ESCs developed abnormal gastruloids surrounded by a cloud of dead cells (Fig. EV3A–D), suggesting a toxic effect of CHIR at these concentrations.

Our data indicate that downregulation of *T-UCstem1* does not affect the ability of ESC aggregates to develop into elongated gastruloids but rather reduces the susceptibility to pharmacological activation of WNT.

### *T-UCstem1* is required for the proper cell type composition in mouse gastruloids

To further investigate the impact of *T-UCstem1* on gastruloid development at a molecular level, we performed a bulk RNA sequencing analysis of Control and *T-UCstem1* KD 5-day-old

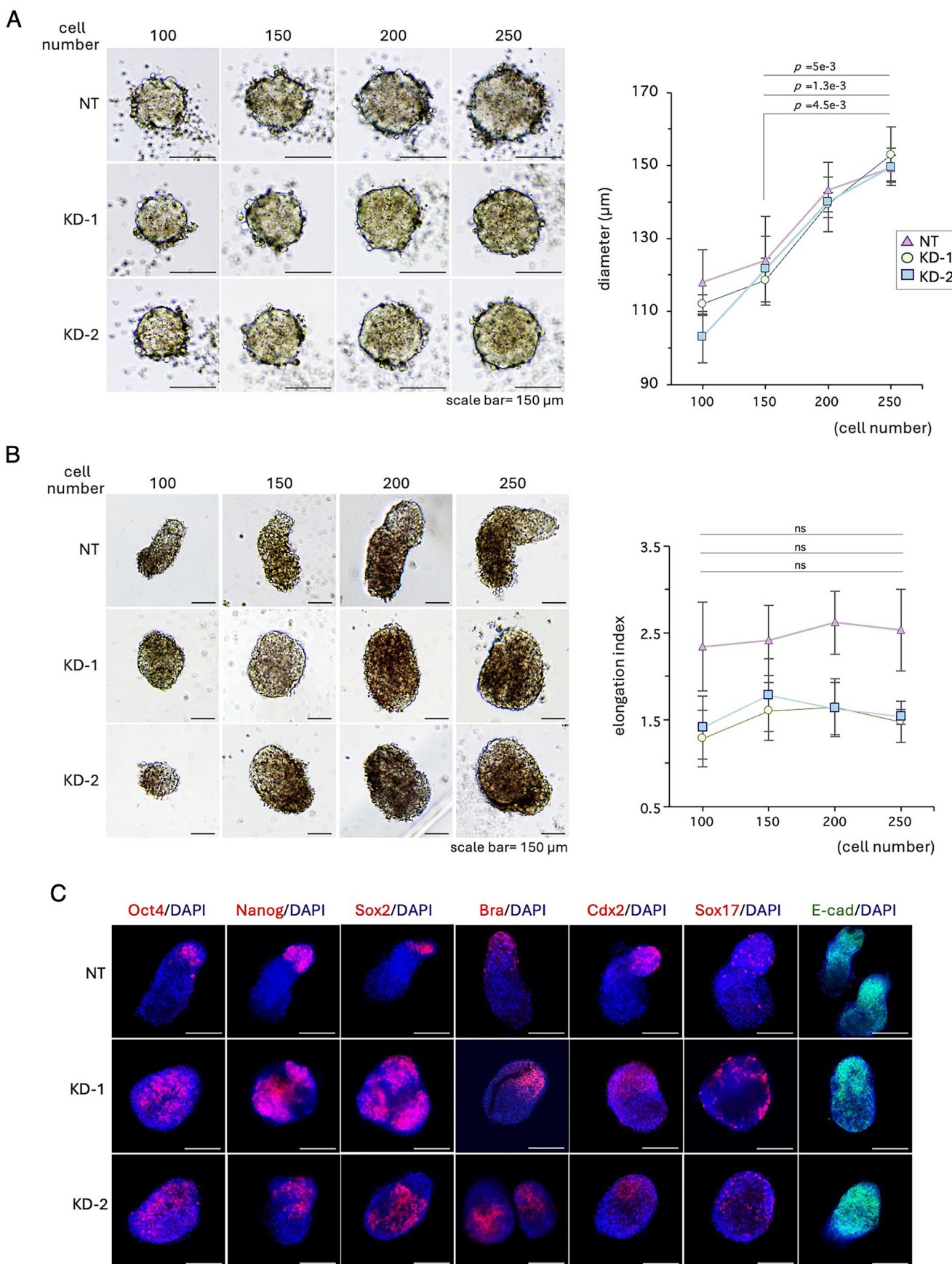

**Figure 3.   *T-UCstem1* KD effect is independent on the initial cell number.**

(A) Representative brightfield images of cell aggregates from 100 to 250 seeded cells (left) and diagrams of aggregate diameter at 48 h (right). Scale bar, 150 μm. Data are shown as mean ± SD ($n = 2$ independent experiments; 10 gastruloids/condition). Statistical significance was assessed by one-way ANOVA with Tukey's multiple comparison test. P values of ≤0.05 were considered statistically significant. (B) Representative brightfield images of gastruloids from 100 to 250 seeded cells (left) and diagrams of gastruloid elongation index at 120 h (right). Scale bar, 150 μm. Data are shown as mean ± SD ($n = 2$ independent experiments; 10 gastruloids/condition). Statistical significance was assessed by one-way ANOVA with Tukey's multiple comparison test. P values of ≤0.05 were considered statistically significant. (C) Representative confocal images of gastruloids from 250 initial cells at 120 h stained with Oct4, Nanog, Sox2 (pluripotency markers), Brachyury, Cdx2, Sox17 (differentiation markers) and E-cad (cell adhesion marker). Nuclei were counterstained with DAPI (blue). Scale bar, 100 μm. All images are representative of three individual gastruloids. Source data are available online for this figure.

gastruloids. The principal component analysis (PCA) revealed significant differences between the two transcriptional profiles with ~6000 differentially regulated genes (fold change >1.5 and fold change <0.667; Padj <0.05) (Figs. 5A and EV4A). Gene ontology analysis unveiled a significant enrichment in terms related to multicellular organism development, locomotion, gastrulation, anteroposterior (A-P) pattern specification, and WNT signalling pathway (Fig. 5B). Particularly, naive and primed pluripotency genes (*Pou3f4, Sox2, Esrrb, Nanog, Tfcp2l1, Tdgf1, Klf2, Zfp42, Klf4, Dnmt3a*) were expressed at higher levels in *T-UCstem1* KD aggregates compared to Control, while A-P pattern specification markers (*Meox1, Foxc2, Tbx18, Foxc1, Pax1, Aldh1a2, Sox17, Fgf8, Gata4, Cdx2*) showed an opposite trend (Fig. 5C). Interestingly, in line with the idea that silencing of *T-UCstem1* reduces the susceptibility of ESC aggregates to WNT signalling, which is crucial for correct gastruloid development (Turner et al, 2017), WNT pathway-related genes were deregulated in *T-UCstem1* KD gastruloids compared to Controls (Figs. 5C and EV4B).

To further investigate this phenotype, and to shed light on the role of *T-UCstem1* on the composition of gastruloids at cellular level, we performed single-cell transcriptomic profiling of Control and *T-UCstem1* KD gastruloids. As expected, in Control gastruloids, most of the cells were fully committed to the three germ layers with an underrepresentation of anterior structures and rostral neuronal fates, and a clear population of neuro-mesodermal progenitors (NMPs) (Suppinger et al, 2023) (Figs. 5D and EV4C). We also observed the emergence of the definitive endoderm lineage, specifically the Anterior Primitive Streak (APS)/Definitive endoderm. Consistent with previous reports, the mesoderm was the most diverse lineage, encompassing cells with somite and paraxial mesodermal identities (Suppinger et al, 2023; van den Brink et al, 2020) (Figs. 5D and EV4C). Interestingly, in *T-UCstem1* KD gastruloids, the cell populations *naive pluripotency, APS/Definitive endoderm* and *caudal epiblast* were statistically more represented, while the cell populations *somite* and *paraxial mesoderm* were more represented in Control gastruloids (Figs. 5E and EV4C). Of note, the *naive pluripotency* labeled population were particularly enriched in *T-UCstem1* KD gastruloids (Figs. 5D and EV4C). The nature and high percentage of cells within the *naive pluripotency* population prompted us to investigate the distribution of cells positive for the three main pluripotency markers: *Oct4, Nanog,* and *Sox2* (Fig. EV4D). Interestingly and consistent with the annotation, while most of the cells expressing the pluripotency markers were localized within the *naive pluripotency* population; a sizable number were also present in other cell populations (e.g., Anterior primitive streak, Caudal epiblast, and NMP), suggesting a potential co-expression of pluripotency and cell commitment markers within these populations (Fig. EV4D).

In line with this hypothesis, we found a significant fraction of *Oct4/Brachyury* double positive cells in *T-UCstem1* KD gastruloids, which was particularly enriched within the *Anterior primitive streak* cell population. Of note, we also found co-expression of *Oct4* with other differentiation markers such as *FoxA2, Nestin, Gata6* and *Sox17* (Figs. 5F and EV4E).

Collectively, these findings demonstrate that the cellular composition of *T-UCstem1* KD gastruloids significantly differs from that of Control gastruloids. Specifically, *T-UCstem1* KD gastruloids were significantly enriched in cell population annotated as *naive pluripotency* as well as those identified as *Anterior primitive streak*, but which actually is composed of cells that exhibit a unique profile integrating molecular features of both pluripotency and differentiation.

### *T-UCstem1* controls the correct cell lineage composition of mouse gastruloids

To further investigate the phenotype of *T-UCstem1* KD gastruloids, we assessed whether the pluripotent cell population enriched in *T-UCstem1* KD gastruloids was also endowed with self-renewal potential. To this end, 4-day-old (96 h AA) Control and *T-UCstem1* KD gastruloids were dissociated, the cells plated at low density (250 cells/cm$^2$) in FBS/LIF/2i medium and the resulting colonies were stained with either crystal violet or alkaline phosphates (AP) and quantified (Fig. 6A,B). Cells derived from Control gastruloids gave rise to few domed-shaped and AP positive colonies; while the same fraction significantly increased (6/7-fold) in the cell population derived from *T-UCstem1* KD aggregates (Fig. 6B). Furthermore, immunofluorescence analysis of the proliferation marker Ki67 showed that the fraction of Ki67-positive cells increased in *T-UCstem1* KD gastruloids compared to Control (Fig. 6C). Of note, the fraction of pluripotent cells was reduced to that of Control in *T-UCstem1* KD1 and KD2 gastruloids treated with higher concentration of CHIR, 5 and 6 μM, respectively. Taken together, these data offer both molecular and functional evidence that *T-UCstem1* controls exit from naive pluripotency of ESC aggregates and suggest that this occurs through modulation of WNT pathway.

We then focused our attention on the cell populations identified by the single-cell analysis, which are characterized by co-expression of pluripotency and differentiation markers. To validate these findings at protein level, 5-day-old (120 h AA) Control and *T-UCstem1* KD gastruloids were dissociated and the cells were double stained with Oct4 and Brachyury. Oct4/Brachyury double positive cells were detected in *T-UCstem1* KD but not in Control gastruloids (Fig. 6D). The concomitant expression of these two key markers indicates the presence of cells displaying molecular feature of both pluripotent and differentiated cells in *T-UCstem1* KD

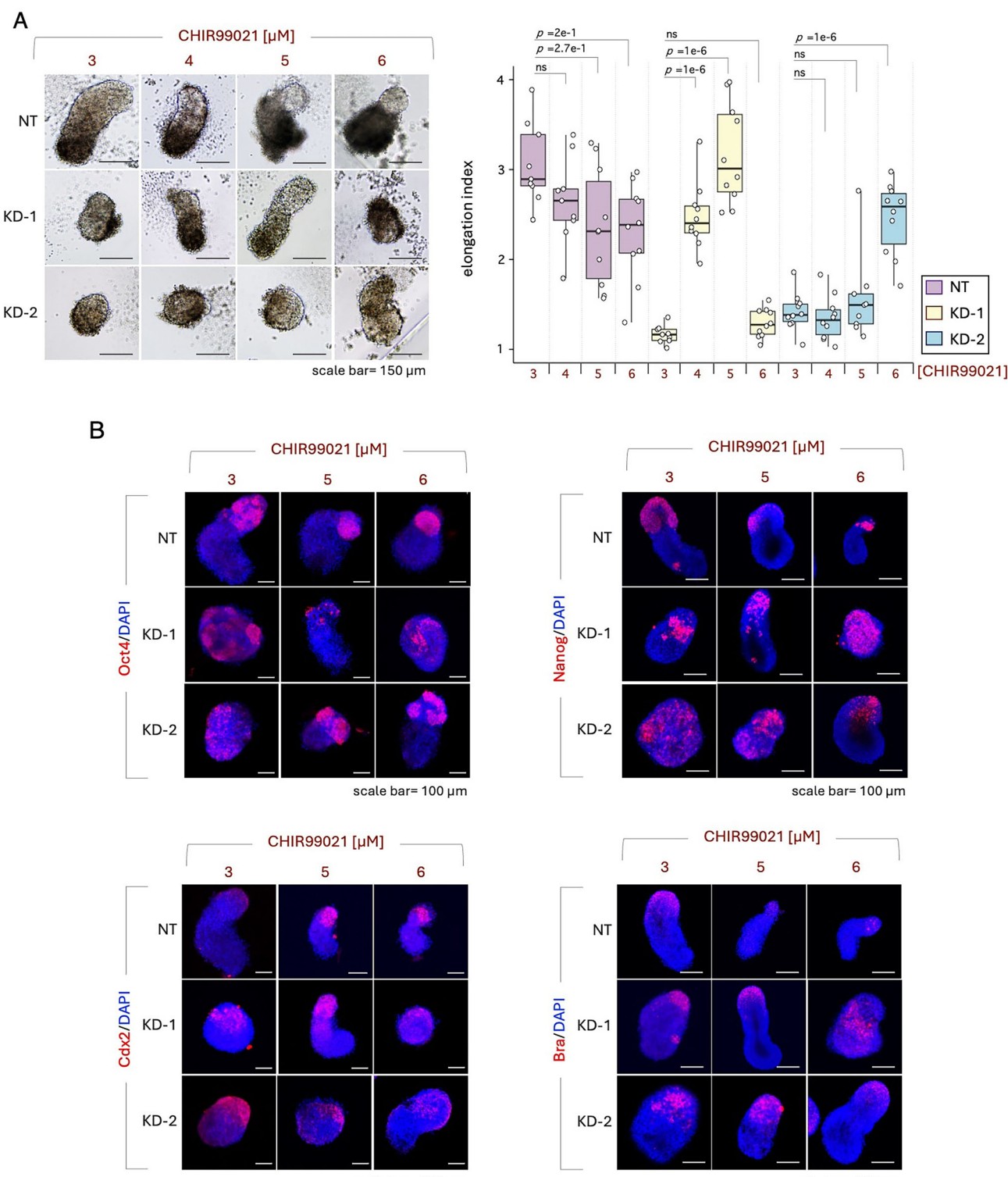

**Figure 4. CHIR99021 rescues *T-UCstem1* KD gastruloid elongation in a dose-dependent manner.**

(A) Dose-response activity of CHIR99021 on gastruloid development. Representative brightfield images of gastruloids treated with increasing concentration of CHIR99021, 3–6 µM (left) and boxplot of gastruloid elongation index at 120 h (right). Scale bar, 150 µm. Data are shown as mean ± SD (*n* = 3 independent experiments; 10 gastruloids/ condition). Boxplots display the minimum, first quartile, median, third quartile, and maximum. Statistical significance was assessed by one-way ANOVA with Tukey's multiple comparison test. *P* values of ≤0.05 were considered statistically significant. (B) Representative confocal images showing the expression of pluripotency (Oct4 and Nanog) and differentiation markers (Cdx2 and Bra) in gastruloids at 120 h, treated with increasing concentrations of CHIR99021 (3–6 µM). Nuclei were counterstained with DAPI (blue). Scale bar, 100 µm. All images are representative of 4 individual gastruloids. Source data are available online for this figure.

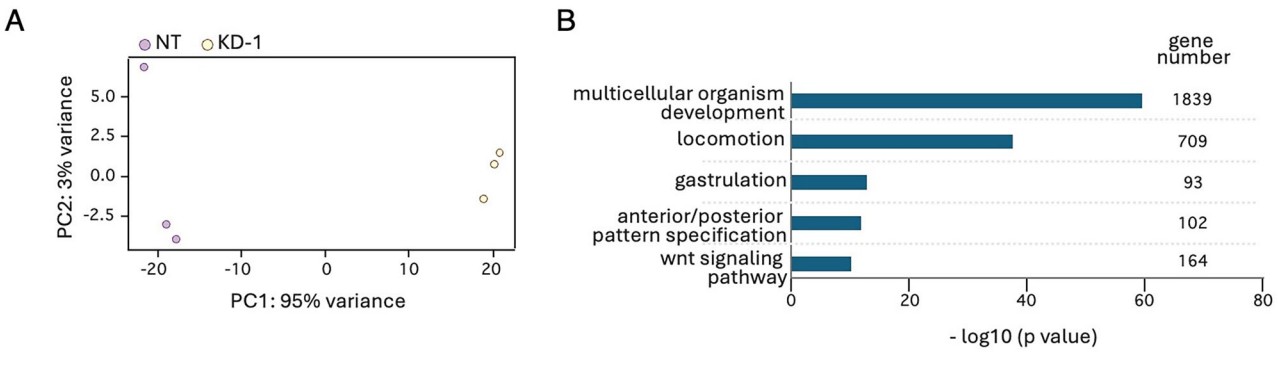

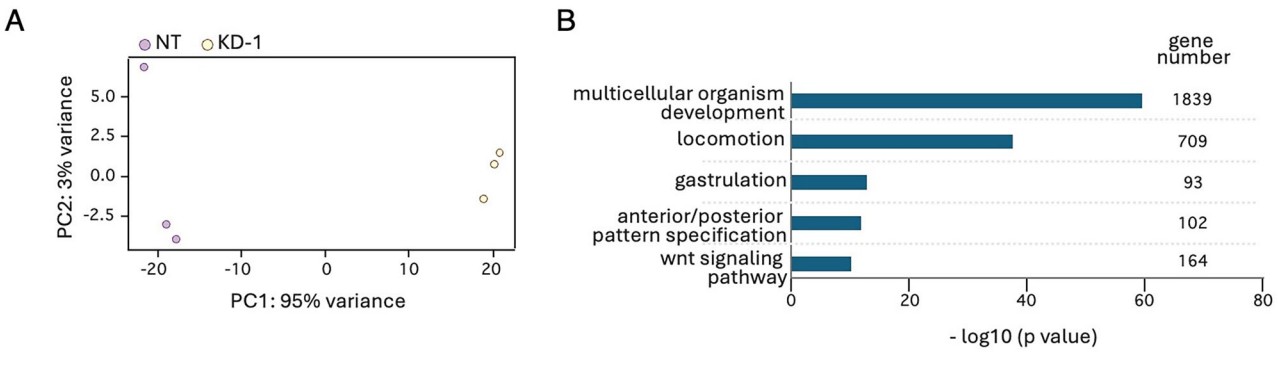

**Figure 5. *T-UCstem1* preserves the gastruloids cellular composition.**

(**A**) Principal-component analysis (PCA) of the expression profiles of Control (NT) and *T-UCstem1* KD-1 gastruloids, treated with 3 µM, at 120 h. (**B**) Enrichment of top five biological process gene ontology terms ordered by *P* value of upregulated genes in *T-UCstem1* KD-1 versus Control (NT) gastruloids. Statistical significance was assessed by Fisher's exact test. *P* values of ≤0.05 were considered statistically significant. (**C**) Heatmaps of upregulated and downregulated genes related to pluripotency, A-P pattern specification and WNT signalling in *T-UCstem1* KD-1 cells versus Control (NT). (**D**) UMAP plot of the subsampled dataset with annotated clusters. (**E**) Odd ratios (OR) of annotated cell populations, in *T-UCstem1* KD-1 vs NT gastruloids. (**F**) UMAP expression plot of Oct4, Brachyury, FoxA2 and Nestin in *T-UCstem1* KD-1 gastruloids. Source data are available online for this figure.

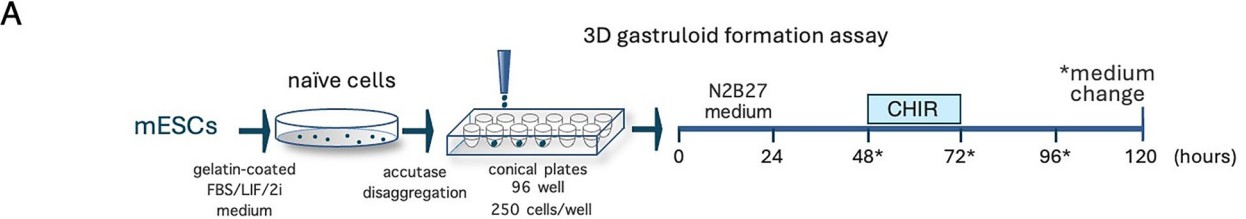

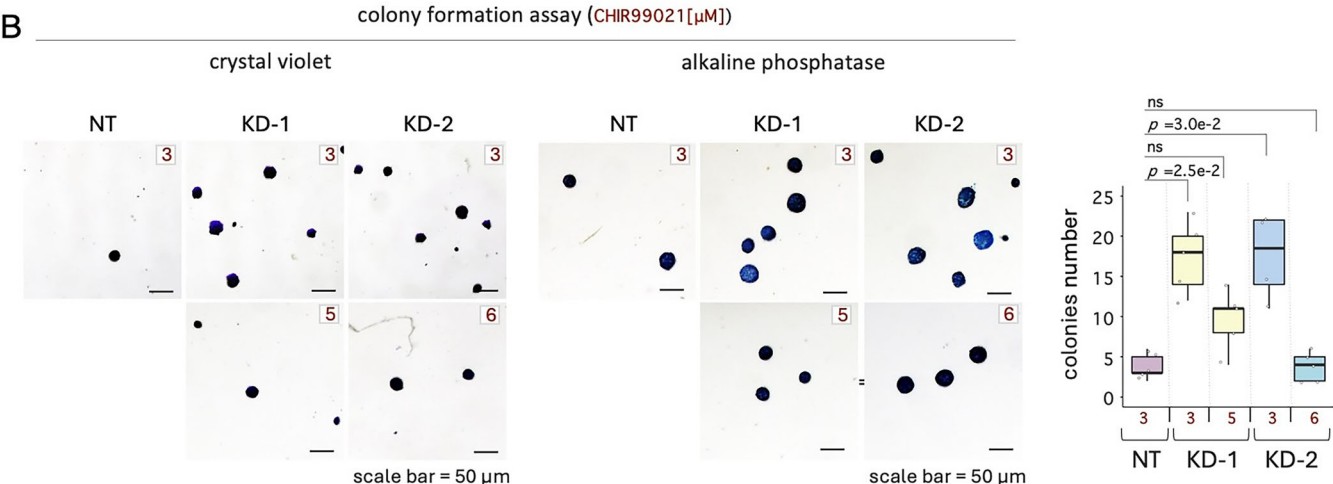

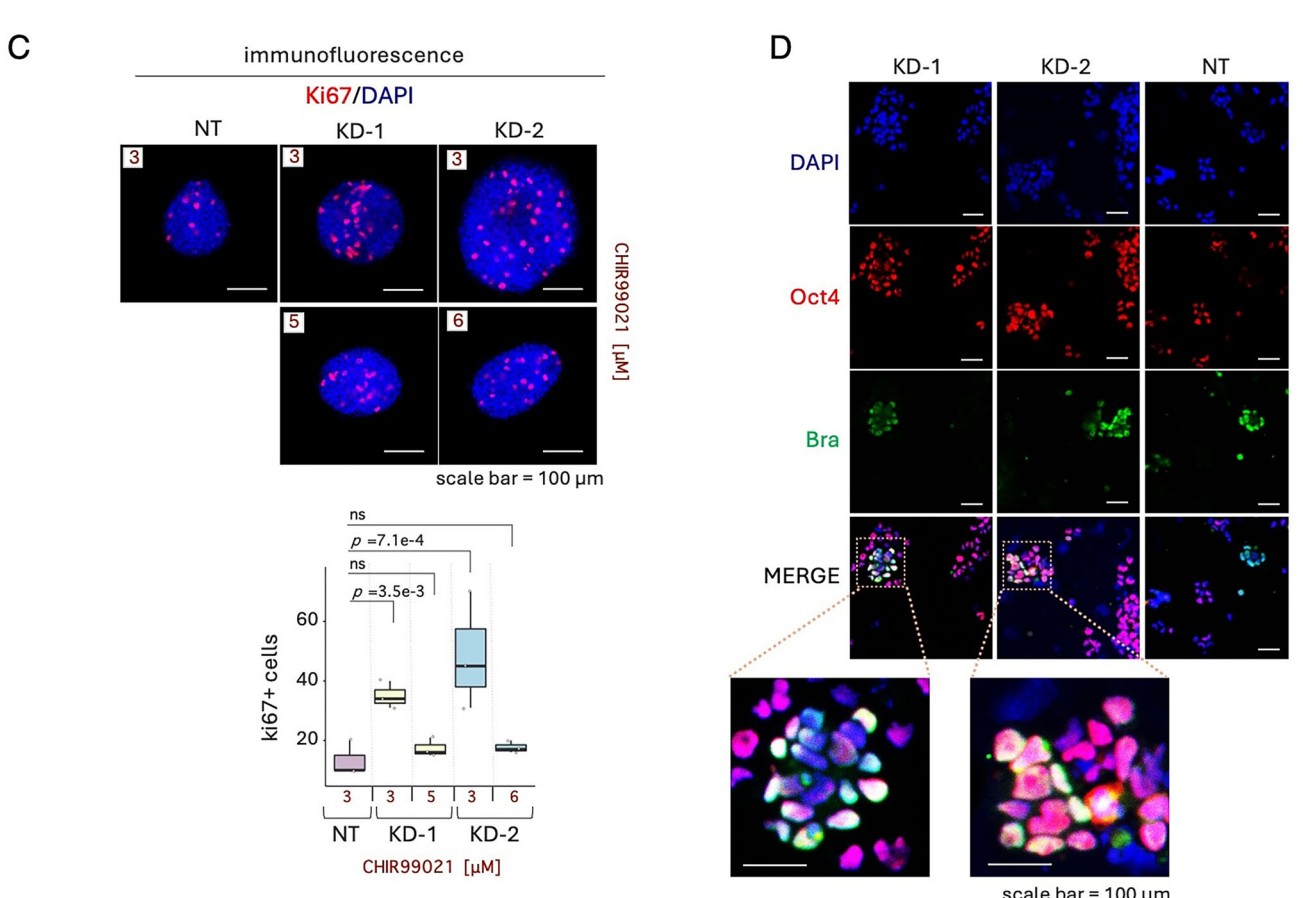

**Figure 6.** *T-UCstem1* is essential for proper cellular exit from pluripotency and establishment of cell identity.

(A) Schematic representation of the experimental design. Control (NT) and *T-UCstem1* KD gastruloids were dissociated by trypsin digestion and the cells plated in FBS/LIF/2i medium at either 250 cells/cm$^2$ for colonies quantification and alkaline phosphatase assay, respectively, after 96 and 120 h in culture. (B) Representative brightfield images (left) of crystal violet and alkaline phosphatase-stained colonies derived from NT and *T-UCstem1* KD dissociated gastruloids. Scale bar, 50 μm. Boxplot diagram (right) showing the number of colonies derived from NT and *T-UCstem1* KD dissociated gastruloids. Data are shown as mean ± SD ($n = 5$ independent experiments; five fields/condition). Boxplots display the minimum, first quartile, median, third quartile, and maximum. Statistical significance was assessed by one-way ANOVA with Tukey's multiple comparison test. $P$ values of ≤0.05 were considered statistically significant. (C) Representative confocal images of Ki67 (proliferation marker) in NT and *T-UCstem1* KD gastruloids at 96 h (top). Scale bar, 100 μm. All images are representative of nine individual gastruloids. Boxplot diagram (bottom) showing the number of Ki67+ cells in NT and *T-UCstem1* KD gastruloids at 96 h. Data are shown as mean ± SD ($n = 3$ independent experiments; three gastruloids/condition). Nuclei were counterstained with DAPI (blue). Boxplots display the minimum, first quartile, median, third quartile, and maximum. Statistical significance was assessed by one-way ANOVA with Tukey's multiple comparison test. $P$ values of ≤0.05 were considered statistically significant. (D) Representative images of Oct4 and Brachyury staining on cytospin NT, *T-UCstem1* KD-1, *T-UCstem1* KD-2 cells derived from the dissociation of gastruloids at 120 h. Nuclei were counterstained with DAPI (blue). Scale bar, 100 μm. All images are representative of five fields/condition. Source data are available online for this figure.

gastruloids, and suggest that *T-UCstem1* exerts a crucial role in the definition of cell lineage specification in developing gastruloids.

## *T-UCstem1* acts through a non-cell-autonomous mechanism that involves DKK1

To investigate the molecular mechanism underlying the function of *T-UCstem1*, we first examined whether it acts through a cell-autonomous or non-cell-autonomous mechanism. To address this issue, we generated chimeric gastruloids by co-culturing Control ESCs with *T-UCstem1* KD-1 and KD-2 ESCs at various ratios, ranging from 4:1 to 1:4 (Fig. 7A–C). Chimeric gastruloids derived from aggregates containing as few as 20% of *T-UCstem1* KD cells already showed defective elongation (Fig. 7B,C), suggesting that *T-UCstem1* acted, at least in part, through a non-cell-autonomous mechanism that can be mediated by soluble factors. To test this hypothesis and considering the reduced sensitivity of *T-UCstem1* KD aggregates to WNT activation, we focused on the WNT pathway-related genes that were deregulated in *T-UCstem1* KD gastruloids as identified by bulk RNA-Seq. Among the most deregulated genes, we found Dickkopf-1 (DKK-1), a secreted inhibitor of beta-catenin-dependent WNT signalling (Fig. 5C). Given this result, we revisited the RNA-Seq data from our previous study and found that DKK-1 was upregulated also in undifferentiated *T-UCstem1* KD ESCs compared to control (Fiorenzano et al, 2018).

Therefore, we validated the RNA-Seq data and analyzed the expression profile of DKK-1 during gastruloid development by real time PCR (Fig. 7D). Interestingly, DKK-1 expression progressively increased in Control gastruloids. In *T-UCstem1* KD gastruloids, this expression profile was conserved, though DKK-1 expression levels

were consistently higher compared to Control (Fig. 7D). Remarkably, single-cell transcriptomic analysis revealed that DKK-1 was upregulated in specific cellular sub-populations, such as the APS/definitive endoderm, which is among the cell populations that are statistically more represented in *T-UCstem1* KD gastruloids compared to Controls (Fig. 7E).

To further explore the hypothesis that *T-UCstem1* acts through DKK-1, we combined pharmacological gain-of-function and loss-of-function approaches by using DKK-1 protein and DKK-1 inhibitor, respectively. Addition of recombinant DKK-1 protein (200 ng/mL) to Control ESC aggregates (Fig. 8A) was able to mimic the phenotype of *T-UCstem1* KD gastruloids at both morphological (Fig. 8B) and molecular (Figs. 8C and EV5A) levels. Notably, DKK-1-treated Control aggregates exhibited reduced elongation index and increased width compared to untreated Controls (Fig. 8D,E), thus supporting the idea that elevated DKK-1 levels explain, at least in part, the mutant phenotype of *T-UCstem1* KD gastruloids. To further explore this hypothesis, we assessed the effect of inhibiting DKK-1 activity in *T-UCstem1* KD-1 gastruloids by using different concentrations of WAY262611, a potent DKK-1 inhibitor and WNT activator (Pelletier et al, 2009). Treatment with WAY262611 (0.1–0.2 μM) rescued in a dose-dependent manner the mutant phenotype of *T-UCstem1* KD aggregates, which developed into properly elongated gastruloids (Figs. 8A,F and EV5B). The rescue was confirmed also at molecular level through expression analysis of pluripotency and differentiation markers (Figs. 8G and EV5C).

Consistent with these findings, real-time PCR analysis of *T-UCstem1* and DKK-1 expression during gastruloid development revealed a complementary pattern, supporting a potential functional correlation between the two molecules (Fig. EV6A). Moreover, overexpression of the ultraconserved element uc.170 in

**Figure 7.** *T-UCstem1* acts through a non-cell-autonomous mechanism.

(A) Schematic representation of the culture protocol: Control (NT) ESCs and *T-UCstem1* KD ESCs were mixed in different ratios at 0 h. CHIR99021 was added between 48 and 72 h; cell aggregates were cultured until 120 h. (B) Representative brightfield images (left) of NT, *T-UCstem1* KD-1 and chimeric gastruloids with different NT:KD-1 ratios (4:1, 3:2, 1:1, 2:3 and 1:4) and boxplot diagram (right) of gastruloid elongation index at 120 h. Scale bar, 150 μm. Data are shown as mean ± SD ($n = 2$ independent experiments; 8 gastruloids/condition). Boxplots display the minimum, first quartile, median, third quartile, and maximum. Statistical significance was assessed by one-way ANOVA with Tukey's multiple comparison test. $P$ values of ≤0.05 were considered statistically significant. (C) Representative brightfield images (left) of NT, KD-2 and chimeric gastruloids with different NT:KD-2 ratios (4:1, 3:2, 1:1, 2:3 and 1:4) and boxplot diagram (right) of gastruloid elongation index at 120 h. Scale bar, 150 μm. Data are shown as mean ± SD ($n = 2$ independent experiments; 8 gastruloids/condition). Boxplots display the minimum, first quartile, median, third quartile, and maximum. Statistical significance was assessed by one-way ANOVA with Tukey's multiple comparison test. $P$ values of ≤0.05 were considered statistically significant. (D) qRT-PCR analysis of DKK-1 expression levels at different stages of control (NT) and *T-UCstem1* KD gastruloid development. Relative RNA level was normalized to *Gapdh*. Data are shown as mean ± SD ($n = 3$ independent experiments). Statistical significance was assessed by one-way ANOVA with Tukey's multiple comparison test. $P$ values of ≤0.05 were considered statistically significant. (E) Violin plot showing DKK-1 expression levels across various sub-populations within NT and KD gastruloids ($n = 698$ cells were analyzed for NT and $n = 692$ cells for KD-1). Source data are available online for this figure.

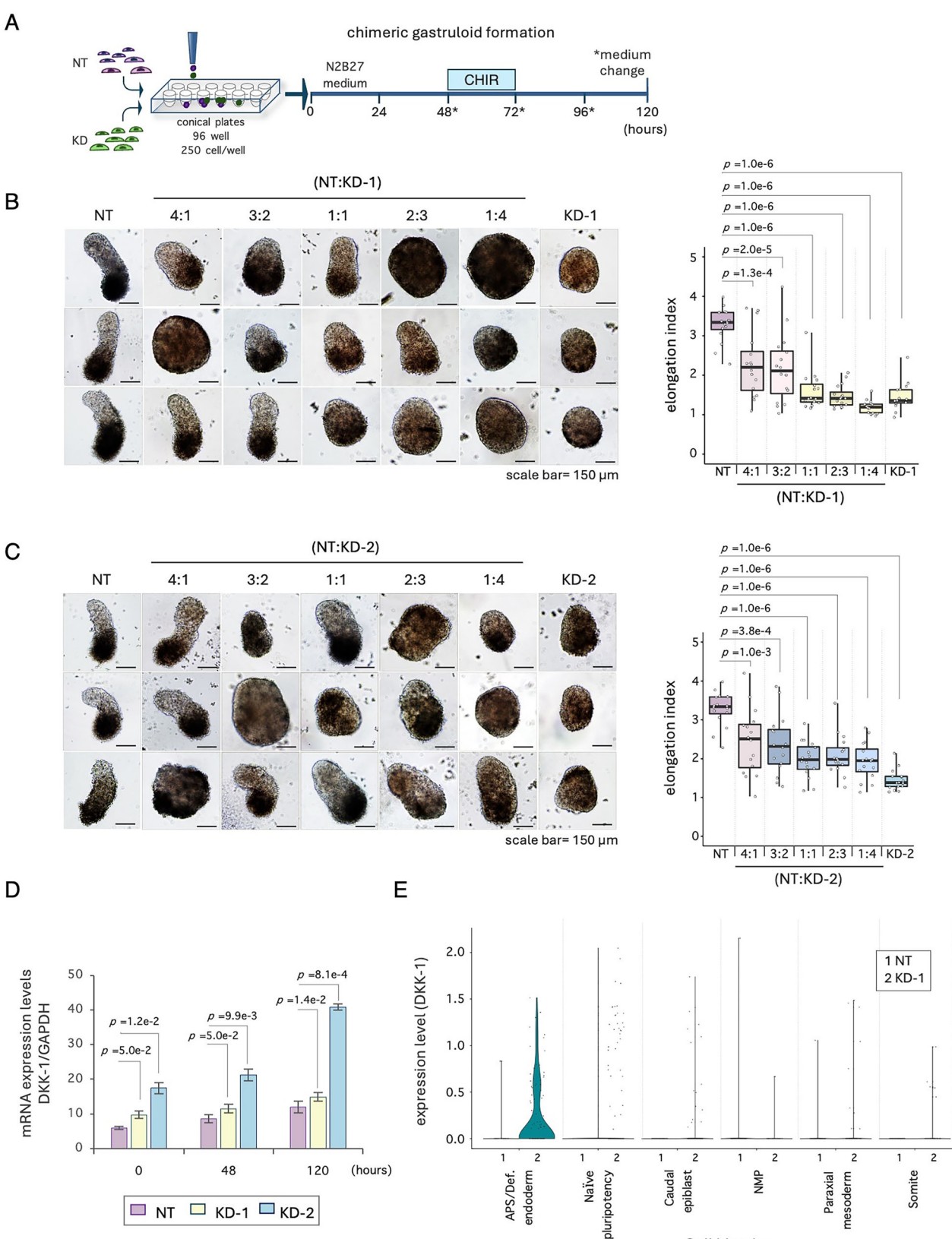

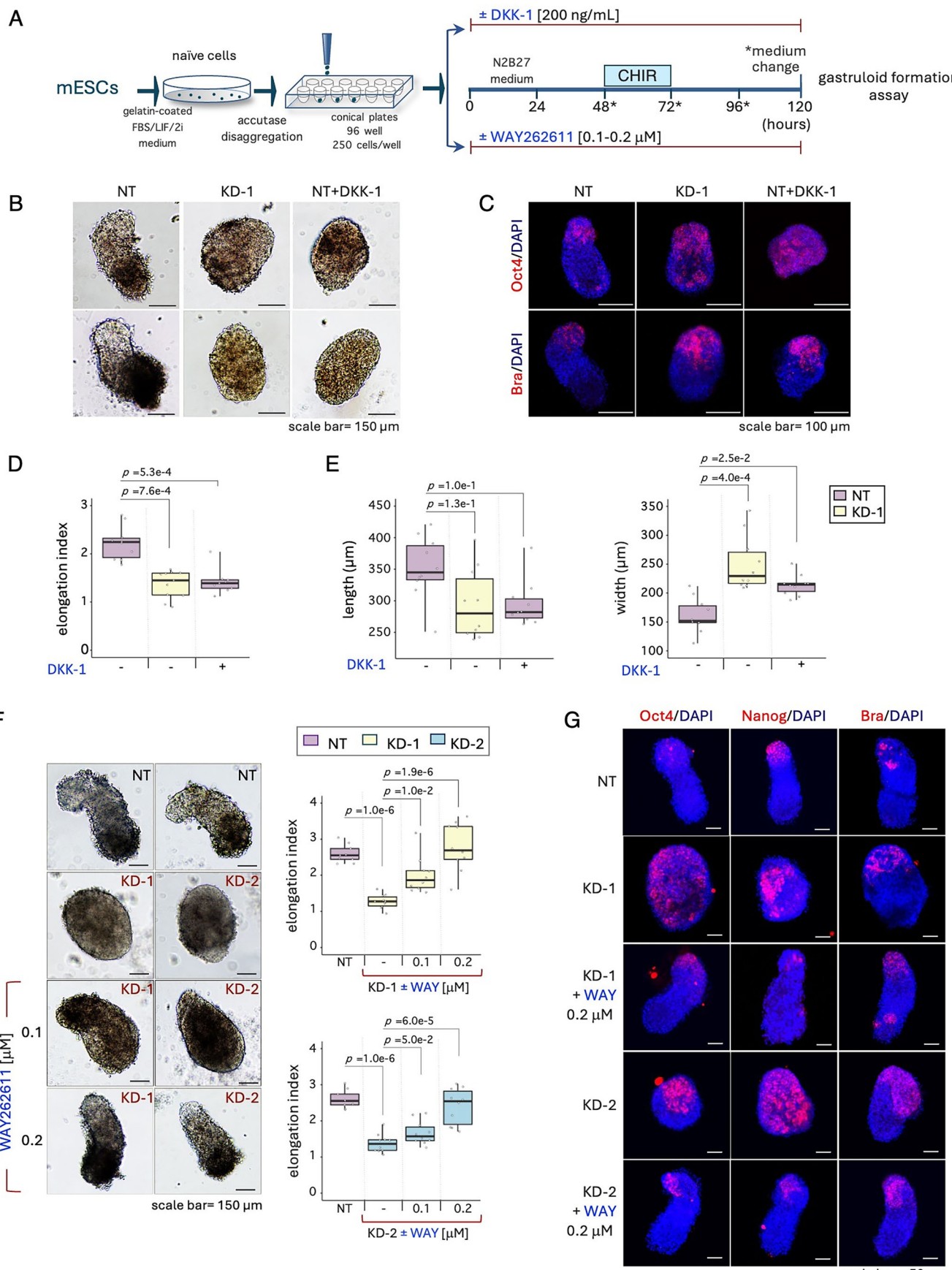

◄ **Figure 8. DKK-1 blocks the correct gastruloid development.**

(A) Schematic representation of the experimental design. DKK-1 and WAY-262611 (DKK-1 inhibitor) were added throughout the entire gastruloid formation assay (from 0 h to 120 h). (B) Representative brightfield images of gastruloids ± DKK-1 recombinant protein (200 ng/mL). Scale bar, 150 µm. (C) Representative confocal images of Oct4 (pluripotency marker) and Brachyury (differentiation marker) in gastruloids ± DKK-1 recombinant protein. Nuclei were counterstained with DAPI (blue). Scale bar, 100 µm. All images are representative of four individual gastruloids. (D) Boxplot diagrams of gastruloids elongation index ± DKK-1 recombinant protein at 120 h. Data are shown as mean ± SD ($n = 3$ independent experiments; 10 gastruloids/condition). Boxplots display the minimum, first quartile, median, third quartile, and maximum. Statistical significance was assessed by one-way ANOVA with Tukey's multiple comparison test. $P$ values of ≤0.05 were considered statistically significant. (E) Boxplot diagrams of gastruloids length and width ± DKK-1 recombinant protein at 120 h. Data are shown as mean ± SD ($n = 3$ independent experiments; 10 gastruloids/condition). Boxplots display the minimum, first quartile, median, third quartile, and maximum. Statistical significance was assessed by one-way ANOVA with Tukey's multiple comparison test. $P$ values of ≤0.05 were considered statistically significant. (F) Dose-dependent effect of WAY262611 on *T-UCstem1* KD aggregates. Representative brightfield images of control (NT) and *T-UCstem1* KD treated with different concentration of DKK-1 inhibitor, WAY262611 (0.1-0.2 µM). Scale bar, 150 µm. Boxplot diagrams of gastruloids elongation index ± WAY262611 at 120 h. Data are shown as mean ± SD ($n = 3$ independent experiments, 10 gastruloids/condition). Boxplots display the minimum, first quartile, median, third quartile, and maximum. Statistical significance was assessed by one-way ANOVA with Tukey's multiple comparison test. $P$ values of ≤0.05 were considered statistically significant. (G) Representative confocal images of Oct4 and Nanog (pluripotency markers) and Brachyury (differentiation marker) in gastruloids NT and *T-UCstem1* KD ± WAY262611 at 0.2 µM. Nuclei were counterstained with DAPI (blue). Scale bar, 50 µm. All images are representative of 4 individual gastruloids. Source data are available online for this figure.

*T-UCstem1* KD clones restored DKK-1 expression to wild-type levels (Figs. 9A and EV6B) and rescued their ability to generate elongated gastruloids, exhibiting morphological and molecular features comparable to those of control cells (Fig. 9B–D).

Altogether, these data demonstrate that *T-UCstem1* regulates gastruloid development through a non-cell-autonomous mechanism and suggest that this occurs, at least in part, through modulation of DKK-1 levels, which in turn determines the sensitivity to WNT signaling activation.

## Discussion

Long non-coding RNAs (lncRNAs) are crucial regulators of gene expression by interacting with target RNAs and proteins in both the cytoplasm and the nucleus (Fico et al, 2019). Despite extensive research, their precise roles in mammalian embryonic development remain largely elusive.

In this study, we demonstrate that *T-UCstem1*, a lncRNA containing an *ultraconserved element* (Fiorenzano et al, 2018), is a key regulator of gastruloid development. Specifically, we provide morphological and molecular evidence that silencing of *T-UCstem1* does not prevent the formation of ESC aggregates, but rather their ability to generate properly elongated gastruloids. Indeed, unlike Control, *T-UCstem1* KD aggregates maintain a spherical/ovoidal morphology with a wide range of shape and size. Immunofluorescence analysis reveals that *T-UCstem1* KD aggregates express Brachyury in a polarized manner, suggesting that symmetry breaking happens in these aggregates. However, they fail to properly elongate. The factors and mechanisms driving elongation in gastruloids remain largely unknown. Recent computational models suggest that a combination of active cellular crawling and differential adhesion might account for gastruloid elongation (de Jong et al, 2024). This is consistent with the increasing evidence that the transition from E- to N-Cadherin is crucial to secure self-organization competence during in vitro gastrulation (Hashmi et al, 2022; Mayran et al, 2023). This aligns with our findings that E-Cadherin expression is maintained in *T-UCstem1* KD gastruloids, which may account at least in part for the failure of proper cell specification and elongation.

We demonstrate that *T-UCstem1* KD gastruloid development can be rescued by increasing the concentration of CHIR from 3 to

5/6 µM. This leads us to conclude that silencing of *T-UCstem1* does not impact the overall competence for gastruloids development but instead reduces the sensitivity of cell aggregates to WNT pathway activation. However, we cannot rule out the possibility that *T-UCstem1* KD cells may exhibit varying sensitivity—either directly or indirectly—to other signaling factors crucial for regulating elongation (McNamara et al, 2023).

Unlike Control, *T-UCstem1* KD gastruloids are enriched in pluripotent cells that maintain *self-renewal* properties. Indeed, both bulk and single-cell transcriptome data support and extend these findings. Specifically, RNA-Seq analysis reveals significant changes in genes involved in pluripotency, locomotion and anteroposterior axis formation. Genes of the WNT signalling pathway are also significantly deregulated in *T-UCstem1* KD gastruloids, which provide molecular support to the idea that silencing of *T-UCstem1* affects the sensitivity to pharmacological activation of WNT.

Transcriptional profiling at the single-cell level highlights significant differences in the cellular composition of *T-UCstem1* KD gastruloids, which are more heterogeneous compared to Control. For instance, *T-UCstem1* KD gastruloids are characterized by the presence of cells that co-express pluripotency and differentiation markers, suggesting that these cells fail to downregulate pluripotency-related genes while attempting to shift toward cell fate commitment. These results, along with the enhanced *self-renewal* ability of cells dissociated from *T-UCstem1* KD gastruloids, lead to the hypothesis that *T-UCstem1* is crucial for the transition from pluripotency to differentiated cell fates.

Mechanistically, we provide evidence that *T-UCstem1* acts non-cell autonomously and suggest that it controls gastruloid patterning and morphogenesis, at least in part, through modulation of the WNT inhibitor, Dickkopf1 (DKK-1). Indeed, the possible link between the lncRNA activity and the cellular response to a morphogen was supported by the transcriptional profile, which reveals higher levels of DKK-1 in *T-UCstem1* KD undifferentiated ESCs and gastruloids compared to Control. Activation of WNT is a crucial event in gastruloid development (Girgin et al, 2021; Turner et al, 2017; van den Brink et al, 2014). Here, we demonstrate that DKK-1 expression levels progressively increase during Control gastruloid formation and are significantly higher in *T-UCstem1* KD gastruloids. These findings align with the idea that WNT pathway must be induced at 48 h AA to break symmetry (Beccari et al, 2018; Turner et al, 2017; van den Brink et al, 2020). However, they also

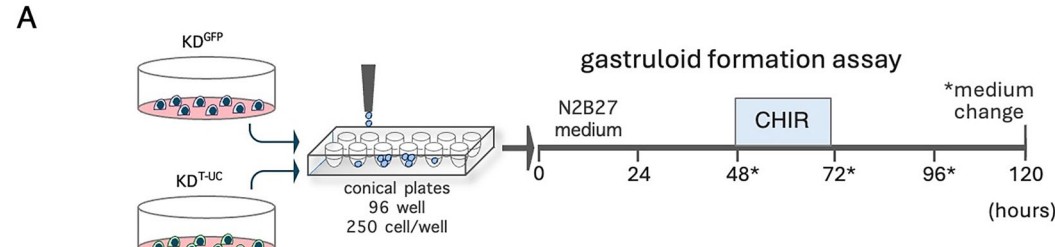

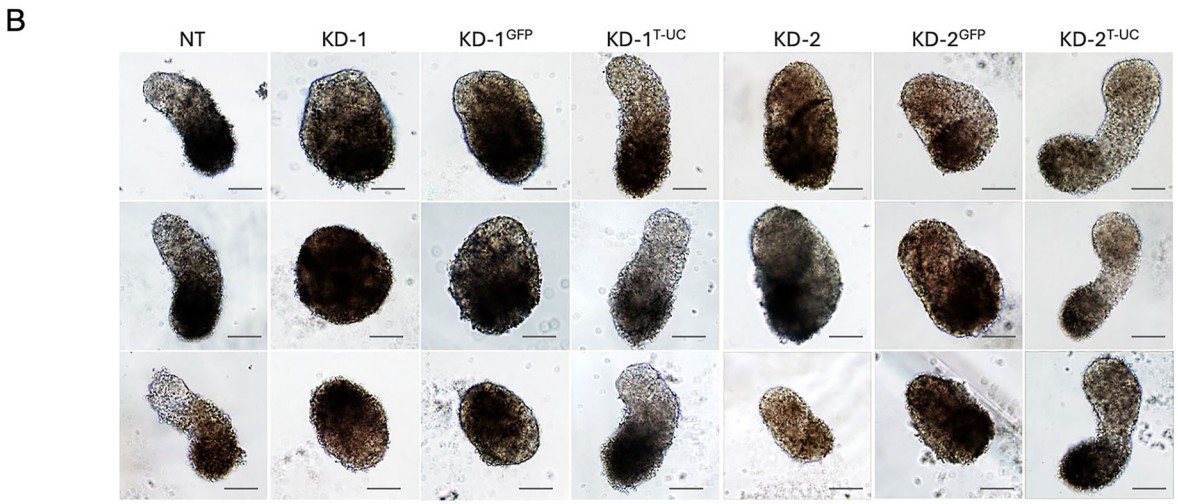

scale bar = 150 µm

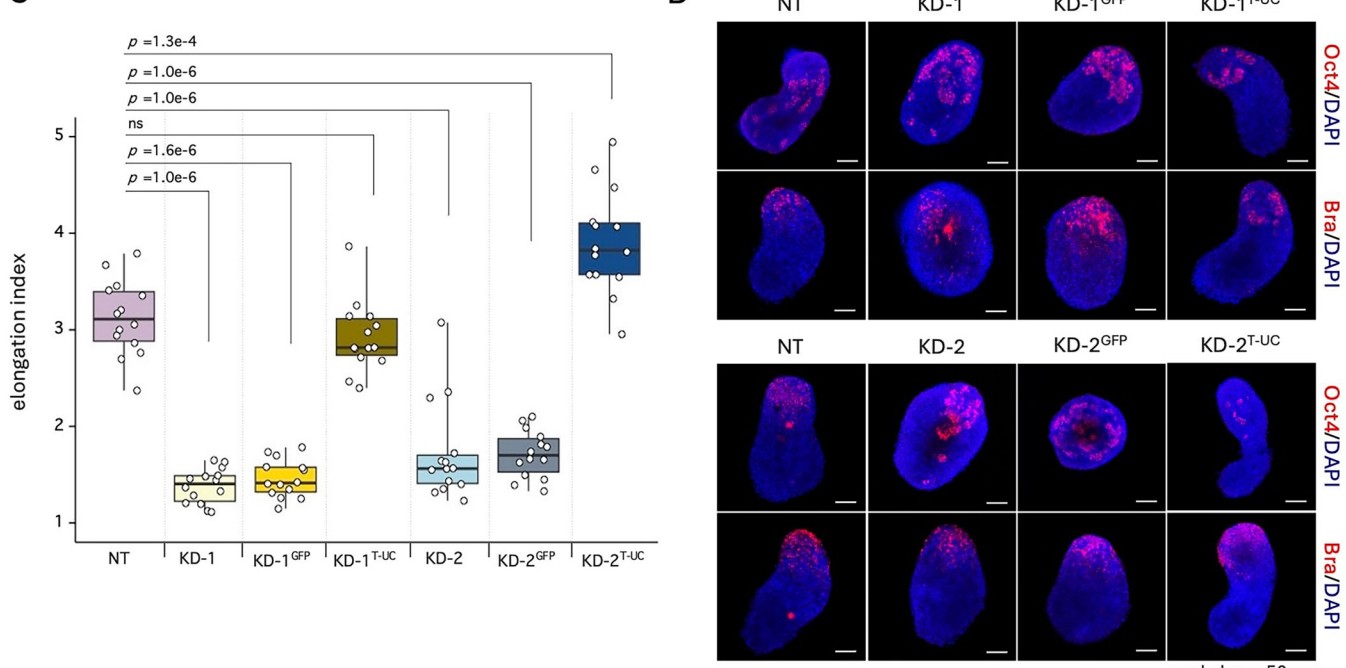

scale bar = 50 µm

**Figure 9. Rescue of gastruloid development by *T-UCstem1* re-expression.**

(A) Schematic representation of the experimental design. *T-UCstem1* KD mESCs overexpressing the ultraconserved element uc.170 (KD[T-UC]) or expressing only GFP as a control (KD[GFP]) were plated on 96-well ultra-low conical plates. CHIR99021 (3 µM) was added between 48 and 72 h; cell aggregates were cultured until 120 h. (B) Representative brightfield images of NT, *T-UCstem1* KD, KD[GFP] and KD[T-UC] gastruloids at 120 h. Scale bar, 150 µm. (C) Boxplot diagrams of gastruloids elongation index at 120 h. Data are shown as mean ± SD ($n = 2$ independent experiments, 14 gastruloids/condition). Boxplots display the minimum, first quartile, median, third quartile, and maximum. Statistical significance was assessed by one-way ANOVA with Tukey's multiple comparison test. $P$ values of ≤0.05 were considered statistically significant. (D) Representative confocal images of Oct4 (pluripotency marker) and Brachyury (differentiation marker) in gastruloids NT, *T-UCstem1* KD, KD[GFP] and KD[T-UC] at 120 h. All images are representative of 4 individual gastruloids. Source data are available online for this figure.

suggest that subsequent silencing of the pathway by DKK-1 is necessary to ensure correct cell differentiation and positioning within the elongated gastruloids.

Of note, the role of DKK-1 in the modulation of gastrulation movements and head formation in vivo is well-known (Caneparo et al, 2007; Mukhopadhyay et al, 2001). Our sc-RNA seq analysis shows that DKK-1 is overexpressed mostly in the subpopulation labelled as *APS/Definitive endoderm*, suggesting that these cells are the main sources of DKK-1 in gastruloids. This observation aligns with the well-defined expression of DKK-1 in the primitive streak and endodermal tissues in vivo (Lewis et al, 2008; Mukhopadhyay et al, 2001). Interestingly, elevated levels of DKK-1 have been found in patients with unexplained recurrent miscarriages (Bao et al, 2013) and in vitro fertilization failure (Koler et al, 2009). However, the effects on pregnancy and embryonic development have not yet been studied in detail, nor analyzed in clinical trials dedicated to this purpose.

At a molecular level, we speculate that *T-UCstem1* regulates DKK-1 expression by stabilizing the polycomb repressive complex, PRC2, at its gene locus, although direct evidence is currently lacking. This hypothesis is based on previous findings that *T-UCstem1* interacts with the PRC2 components EZH2 and SUZ12 in undifferentiated ESCs (Fiorenzano et al, 2018), and the fact that DKK-1 is a PRC2-targeted bivalent gene (Studach et al, 2012).

In conclusion, our study provides unprecedented evidence that the fine regulation of gastruloid development also involves ncRNAs and their activities. Specifically, *T-UCstem1* orchestrates the differentiation and spatial organization of different cell populations within the gastruloid by modulating sensitivity to the WNT-β-catenin signalling pathway, at least in part through the regulation of DKK-1. These findings not only deepen our understanding of the molecular mechanisms governing mammalian embryogenesis, and open the way to investigate the implications of lncRNA dysregulation in developmental and reproductive disorders.

# Methods

### Reagents and tools table

| Reagent/resource | Reference or source | Identifier or catalog number |
|---|---|---|
| **Experimental models** | | |
| Non-targeted ESCs | Fiorenzano et al, 2018 | N/A |
| *T-UCstem1* KD-1 ESCs | Fiorenzano et al, 2018 | N/A |
| *T-UCstem1* KD-2 ESCs | Fiorenzano et al, 2018 | N/A |

| Reagent/resource | Reference or source | Identifier or catalog number |
|---|---|---|
| **Recombinant DNA** | | |
| p2xCAGS-GFP | Pascale et al, 2020 | N/A |
| p2xCAGS-UC | Pascale et al, 2020 | N/A |
| **Antibodies** | | |
| Oct4 | Santa Cruz | Cat#8628 |
| Nanog | Cell Signaling | Cat#88225 |
| Sox2 | Cell Signaling | Cat#D1C7J |
| Brachyury | Cell Signaling | Cat#D2Z3J |
| Cdx2 | Cell Signaling | Cat#39775 |
| Sox17 | R&D | Cat#AF1924 |
| Nestin | Santa Cruz | Cat#33677 |
| E-cadherin | Takara bio | Cat#M108 |
| Ki67 | Thermo Fisher | Cat#14520 |
| Donkey Anti Rabbit 594 | Invitrogen | Cat#A21207 |
| Donkey Anti Goat 594 | Invitrogen | Cat#A11058 |
| Donkey Anti Goat 488 | Invitrogen | Cat#A11055 |
| **Oligonucleotides and other sequence-based reagents** | | |
| DKK-1 | fw oligo: atgtagggctgctggactgt | N/A |
| | rv oligo: gtgggcatcatggtttacgg | N/A |
| GADPH | fw oligo: tgcaccaccaactgcttagc | N/A |
| | rv oligo: tcttctgggtggcagtgatg | N/A |
| T-UCstem1 | fw oligo: tgagtctttgcctctctttgg | N/A |
| | rv oligo: aagtgctgaagcacccctta | N/A |
| U6 | fw oligo: ctcgcttcggcagcacatat | N/A |
| | rv oligo: aacgcttcacgaatttgcgt | N/A |
| **Chemicals, enzymes and other reagents** | | |
| DMEM | Gibco | Cat#41965-039 |
| FBS- fetal bovine serum | Gibco | Cat#16141-079 |
| L-Glutamine (200 mM) | Gibco | Cat#25030-024 |
| Penicillin/Streptomycin | Gibco | Cat#15140-122 |
| Sodium Pyruvate (100 mM) | Gibco | Cat#11360-039 |

| Reagent/resource | Reference or source | Identifier or catalog number |
|---|---|---|
| 2-Mercapto-Ethanol | Sigma-Merck | Cat#M-7522 |
| Leukemia inhibitory factor | Sigma-Merck | Cat#ESG1107 |
| Gelatin | Sigma-Merck | Cat#G7765 |
| Trypsin | Gibco | Cat#25300-054 |
| DPBS | Gibco | Cat#14190-144 |
| DMEM F12 | Gibco | Cat#21331-020 |
| Neurobasal | Gibco | Cat#21103-049 |
| B27 Supplement | Gibco | Cat#17504044 |
| N2 Supplement | Gibco | Cat#17502048 |
| CHIR99021 | Selleckchem | Cat#S1263 |
| PD0325091 | Selleckchem | Cat#S1036 |
| DKK-1 recombinant protein | R&D systems | Cat#5439 |
| WAY262611 | Sigma-Aldrich | Cat#317700 |
| Accutase | Merck | Cat#A6964 |
| G418 | InvivoGen | Cat#ant-gn-1 |
| TruSeq library preparation kit | Illumina | N/A |
| NextSeq 500 High Output kit | Illumina | N/A |
| Chromium Single Cell 3' reagent kit | 10X Genomics | N/A |
| SPRIselect magnetic beads | Beckman Coulter | Cat#B23319 |
| Qubit dsDNA BR Assay | Thermo Fisher Scientific | Cat#Q32850 |
| D1000 screentape on the TapeStation 4150 | Agilent Technologies | Cat#G2992AA |
| **Software** | | |
| Fiji | ImageJ | |
| R Studio | https://www.rstudio.com | |
| David, v2023q3 | https://david.ncifcrf.gov/ | |
| Cell Ranger Single-Cell, v.8.0 | 10x Genomics | |
| **Other** | | |
| DMI6000B microscope | Leica Microsystems | |
| AF6000 | Leica Microsystems | |
| Nikon A1 confocal microscope | Nikon | |
| Shandon Cytocentrifuge (CytoSpinTM 4) | Thermo Fisher Scientific | |
| Illumina NextSeq550 System | Illumina | |
| Chromium Controller | 10X Genomics | |

## Conservation analysis

The conservation of *T-UCstem1* in vertebrates has been obtained from UCSC Genome browser. Uc.170 track was highlighted together with the Multiz Alignments of 35 Vertebrates and Pairwise alignments of some species. Conservation has been calculated by phyloP algorithm (Siepel et al, 2005) which can measure acceleration (faster evolution than expected under neutral drift) as well as conservation (slower than expected evolution). Note that excluding species from the pairwise display does not alter the conservation score display.

## Cell culture and treatments

Mouse embryonic stem cells (ESCs) used in this study included non-targeting control cells (NT) and two independent T-UCstem1 knockdown clones (designated KD-1 and KD-2), all derived from the TBV2 129/SvP line as previously described (Fiorenzano et al, 2018). These specific cell lines were originally generated and characterized in detail in the aforementioned study, where the knockdown (KD) efficiency of T-UCstem1 was thoroughly validated. The KD-1 and KD-2 clones were selected based on their differing levels of T-UCstem1 silencing, enabling the investigation of potential dose-dependent effects on ESC behavior.

ESCs were grown at high density in DMEM/FBS/LIF as previously described (Fiorenzano et al, 2016). All the experiments were performed between the 10th and the 25th cell passage. For all cell lines, Mycoplasma-free state is routinely (twins/year) tested by PCR-based assay. All the reagents used for cell culture and treatments are reported in Reagents and tools Table.

## Colony formation assay

TBV2 mESC lines were grown using the protocol described by Cermola et al (Cermola et al, 2021). Briefly, to perform colony formation assay, feeder cells and ESCs were pre-plated to allow the quick attachment of the feeder cells in order to get a pure ESC population. ESCs were seeded at low density (250 cells/cm$^2$) on gelatin-coated plates in DMEM/FBS containing a mix of three naive pluripotency-inducing cytokines/inhibitors: Leukemia Inhibitor Factor (LIF; 1000 u/mL, ESGRO-Millipore), a WNT signalling agonist (CHIR99021; 3 μM, Selleckchem), and a MEK inhibitor (PD0325091; 1 μM, Selleckchem). At 72 h, the medium was refreshed. After 5 days, colonies were fixed in 4% paraformaldehyde (PFA) and stained with crystal violet and alkaline phosphatase as previously described (D'Aniello et al, 2015). Images were collected on a DMI6000B microscope (Leica Microsystems). The morphological classification (domed/flat) was performed blinded by two investigators.

## Plasmids and cells electroporation procedure

For the T-UCstem1 expression rescue experiments, the plasmids 2xCAGS-GFP and 2xCAGS-UC, previously described by Pascale et al (Pascale et al, 2020) were used.

For electroporation the mESCs were dissociated using 0.05% trypsin-EDTA and resuspended in 700 μL of Phosphate Buffered saline (PBS 1X; Gibco; cat: 14190-144). Approximately, $1 \times 10^6$ cells were transfected with 20 μg of linearized plasmid DNA and transferred into electroporation cuvette (Bio-Rad; cat: 165-2088). Electroporation was performed using a Bio-Rad Gene Pulser II Electroporation System Capacitance Extender Plus set to 250 V and 500 μF. Following electroporation, cells were seeded on plates and the selection of clones was initiated 48 h later by adding Neomycin (G-418) at 200 μg/mL (InvivoGen; cat: ANT-GN-1). Neomycin-resistant cell pools were either harvested for RNA extraction and for gastruloids formation assay.

## Gastruloid formation assay and analysis

Gastruloid formation assay was performed as previously described (Baillie-Johnson et al, 2015; Cermola et al, 2021). Naive ESCs were cultured at low density (250 cells/cm$^2$) and the resulting colonies were dissociated with accutase (Sigma-Aldrich; cat: A6964; 3 min at 37 °C). Cells were seeded in N2B27 at $1.0–2.5 \times 10^2$ cells/well (40 µL) in ultra-low attachment 96 multi-well plates (Corning Costar; cat: 7007) and allowed to aggregate. At 48 h AA, CHIR was added (3 µM) to the culture medium and maintained for 24 h. From 72 h onward, the medium (150 µL) was refreshed daily up to 120 h. DKK-1 recombinant protein (R&D systems, cat: 5439-10 µg) was dissolved in PBS 1X containing at least 0.1% BSA at a concentration of 100 µg/mL and used at a concentration of 200 ng/mL. WAY262611 (Sigma-Aldrich, cat: 317700-10 mg) was dissolved in DMSO at a concentration of 25 mg/mL and used at 0.1 and 0.2 µM. Both reagents were added during throughout the assay. Gastruloids were imaged using EVOS. The aggregate's diameter at 48 h was analyzed using ImageJ 1.46r software (https://imagej.nih.gov/ij). The elongation index was calculated using ImageJ-Fiji (BIOP plugin). All the reagents used for the gastruloids formation assay are reported in Reagents and tools Table.

## Immunofluorescence on gastruloids

To perform immunofluorescence on gastruloids, we used the protocol previously described (Baillie-Johnson et al, 2015; Cermola et al, 2021). Briefly, gastruloids were washed (3×, 10 min) at RT with PBS, then with PBSFT (PBS/10% FBS/0.5% Triton X) (3×, 10 min) and finally with PBSFT (1 h, 4 °C). Gastruloids were then incubate with the following specific antibodies (24–48 h, at 4 °C) on a low-speed orbital rocker. Images were obtained using confocal Nikon A1 microscope. Then NIS Element C (Nikon, Tokyo, Japan) software was used for image acquisition/ elaboration. Primary and secondary antibodies are listed in Reagents and tools Table.

## Cytospin and immunofluorescence analysis

ESCs were dissociated with trypsin-EDTA (5 min at 37 °C), cells were resuspended in 15% FBS, PBS 1× and centrifuged (800 rpm for 8 min) onto glass slides ($5 \times 10^5$ cells/spot) using a Thermo Shandon Cytocentrifuge (CytoSpinTM 4). Cytospin cell samples were fixed in 4% paraformaldehyde (PFA), permeabilized in 0.1% Triton X-100 (10 min at RT) and incubated overnight at 4 °C with primary antibodies. After washing (0.5% Tween-1x PBS), cells were incubated with appropriate secondary antibodies. Nuclei were counterstained with DAPI (Invitrogen). Images were obtained using the DMI6000B microscope and the DFC 350FX B/W digital camera (Leica Microsystems). Confocal images were obtained on a Nikon A1 microscope. The AF6000 (Leica Microsystems) and NIS Element C (Nikon, Tokyo) software were used for image acquisition/elaboration. Primary and secondary antibodies are listed in Reagents and tools Table.

## RNA extraction and real-time PCR

To perform RNA extraction, we used Trizol (Thermo Fisher Scientific, Invitrogen; cat: 15596018). Total RNAs were reverse transcribed with the High-Capacity cDNA Reverse Transcription Kit (Thermo Fisher Scientific; cat:4368814). To analyze gene expression, we performed real-time PCR using using SYBR Green PCR master mix (FluoCycle II SYBR, EuroClone). The primer sequences are reported in Reagents and tools Table.

## Bioinformatics analysis of bulk RNA sequencing data

cDNA libraries were prepared using the Illumina TruSeq library preparation kit and sequenced with $2 \times 150$ bp paired end reads on an Illumina NextSeq 500 High Output kit. Raw base calls were demultiplexed and converted into sample specific fastq format files using default parameters of the bcl2fastq program provided by Illumina (v2.19). Resulting fastq files were quantified using Salmon (v1.1) using Gencode version 33 as the gene model. Downstream analysis was performed in R (v4.2.2) using DESeq2 (v1.38.3) for differential expression analysis and EnhancedVolcano (1.16) for plotting volcano plots. Pathway analysis was performed using the KEGG database (ref mmu04310) and rendered with pathview (v1.38.0). Gene Ontology was performed using the David software (v2023q3) (https://david.ncifcrf.gov/). Statistical significance was assessed by Fisher's exact test. $P$ values of ≤0.05 were considered statistically significant.

## Single-cell RNA-seq and data analysis

For scRNA-seq, gastruloids from NT and *T-UCstem1* KD-1 were generated as indicated above, collected after 120 h and processed using the Chromium Next GEM Single Cell 3′ kit v3.1 (10X Genomics, Pleasanton, CA), according to manufacturer's instructions. Briefly, gastruloids were collected in 15 mL tubes, pelleted by gravity and dissociated by trypsin incubation at 37 °C for 7 min, washed with PBS and resuspended in cold PBS supplemented with 0.1% BSA. The single-cell suspensions were then visually inspected for determining cell number and viability as well as the absence of residual cellular aggregates and finally loaded into the Chromium Controller (10X Genomics) to generate single cell GEMs. After GEM generation, all the steps of cDNA synthesis, clean-up and amplification, DNA fragmentation, end-repair, adapter ligation and indexing PCR-amplification were done with the Chromium Single Cell 3' reagents following the manufacturer's protocol while the DNA purification as well as the size selection steps were done with SPRIselect magnetic beads (Beckman Coulter, Brea, CA) as indicated by the 10X protocol. Finally, the concentration of each indexed library was determined by using Qubit dsDNA BR Assay (Thermo Fisher Scientific, USA) and their quality were assessed with the D1000 screentape on the TapeStation 4150 (Agilent Technologies, Santa Clara, CA). In the end, an equimolar amount of both the DNA libraries was pooled together and subject to cluster generation and sequencing into the Illumina NextSeq550 System (Illumina, San Diego, CA) in a paired-end dual index format with the following sequencing cycles: 28 for Read1, 10 for Index i7, 10 for Index i5 and 90 for Read2.

The Cell Ranger Single-Cell software (v.8.0, 10x Genomics) was used for processing data as according to manufacturer. Briefly, reads will be first assigned to cells and then aligned to the mouse genome (refdata-gex-GRCm39-2024-A, GRCm39) using STAR(-Dobin et al, 2013). The output filtered gene expression matrices were analyzed by R software (R Core Team, 2023) with the Seurat package (v.5) (Hao et al, 2024). Low-quality cells with a high

percentage (>15%) of reads from mitochondrial genes and with few or too many genes removed from further analyses. Seurat v5 standard analysis workflow run followed by integration of the two samples with rpca reduction. Optimization of cluster parameters and annotation according to ScType algorithm (Ianevski et al, 2022) given a list of 19 different gastruloids cell population markers from Suppinger et al, 2023 (Suppinger et al, 2023) establishing an annotation framework to allow for a more complete and explorative analysis. Labelling step resulted in 6 cell population labeled with embryos or gastruloids signatures. The strength of the signature can be retrieved in the Dataset EV1 file. After the downsampling of the NT gastruloids (to easy the visual comparison with KD1), the uniform manifold approximation and projection (UMAP) algorithm used to depict the cell populations and the expression of specific makers. For each cell population Fisher test performed to pinpoint the significant odd ratios (OR) (FDR-adjusted $P$ value < 0.01) in KD-1 vs NT gastruloids.

## Statistical analysis

The number of independent experiments is reported as "$n$" in the caption of figure and the total sample is indicated. Statistical significance was assessed by one-way ANOVA with Tukey's multiple comparison test. $P$ values of ≤0.05 were considered statistically significant. Results are presented as mean ± SD or as a boxplot/dot plot displaying the minimum, first quartile, median, third quartile, and maximum. Boxplot were generated using RSstudio software (https://www.rstudio.com/).

Morphometric analysis of gastruloids, quantitative analyses of ESC colony formation and Ki67-positive cells by immunofluorescence, were performed by two independent experimenters.

## Data availability

Raw RNA-seq data generated in this study have been deposited in the Gene Expression Omnibus (GEO) database with accession code GSE278442. Raw sc-RNA-seq data generated in this study have been deposited in the functional genomics data collection (ArrayExpress), with accession code E-MTAB-14503.

The source data of this paper are collected in the following database record: biostudies:S-SCDT-10_1038-S44318-025-00558-2.

## Peer review information

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

## Acknowledgements

We are grateful to Salvatore Arbucci and members of the Integrated Microscopy and FACS Facilities of IGB-ABT, CNR. This study was supported by NUTRIAGE-CNR (project FOE-2021 DBA.AD005.225), INV-ACT-CNR (project

FOE-2022), Italian Ministry of University and Research (M.U.R) Project PRIN P2022LTFLR funded by Next Generation EU National Recovery and Resilience Plan (PNRR), and Next Generation EU Ministry of Health National Recovery and Resilience Plan (PNRR), Project AnIMANDo (PNRR-MCNT2-2023-12377937) and MNESYS (PE0000006) to A Fico and Italian Ministry of University and Research (M.U.R) Project  PRIN P20224ZY5P funded by Next Generation EU National Recovery and Resilience Plan (PNRR) to G Minchiotti. A Fiorenzano acknowledges funding through the Swedish Research Council (2022-01432).

## Author contributions

**Arianna Coppola**: Conceptualization; Data curation; Formal analysis; Methodology; Conceived and designed research, performed research, collected data and analyzed data, analyzed the bulk transcriptome sequencing. **Filomena Amoroso**: Data curation; Methodology; Performed research, collected data and analyzed data. **Federica Saracino**: Data curation; Methodology; Performed research, collected data and analyzed data. **Gennaro Andolfi**: Methodology; Performed research, collected data and analyzed data. **Edoardo Sozzi**: Data curation; Software; Formal analysis; Methodology; Analyzed the bulk transcriptome sequencing. **Paolo Salerno**: Methodology; Performed and analyzed the single cell transcriptomic. **Pietro Zoppoli**: Data curation; Software; Formal analysis; Performed and analyzed the single cell transcriptomic. **Alessandro Fiorenzano**: Formal analysis; Supervision; Funding acquisition; Analyzed the bulk transcriptome sequencing. **Giuseppe Merla**: Supervision; Performed and analyzed the single cell transcriptomic. **Eduardo Jorge Patriarca**: Conceptualization; Formal analysis; Conceived and designed research, analyzed the bulk transcriptome sequencing. **Gabriella Minchiotti**: Funding acquisition; Writing—review and editing; Gave conceptual advice and contributed to the editing of the manuscript. **Annalisa Fico**: Conceptualization; Supervision; Funding acquisition; Writing—original draft; Project administration; Conceived and designed research, wrote the manuscript.

Source data underlying figure panels in this paper may have individual authorship assigned. Where available, figure panel/source data authorship is listed in the following database record: biostudies:S-SCDT-10_1038-S44318-025-00558-2.

## Disclosure and competing interests statement

The authors declare no competing interests.

# Expanded View Figures

**Figure EV1.   *T-UCstem1* conservation across species and shape heterogeneity showed by *T-UCstem1* KD aggregates.**

(**A**) In red NM_1138112 is *T-UCstem1*, while uc.170 is the ultra-conserved sequence by Bejerano et al, 2004 *Cons 35 verts* is the phyloP plot, with sites predicted to be conserved are assigned positive scores (shown in blue), while sites predicted to be fast evolving are assigned negative scores (shown in red). Multiz Alignments of 35 Vertebrates depicts pairwise alignments of some species to the mouse genome indicating alignment quality. (**B**) The figure showing the heterogeneity of *T-UCstem1* KD gastruloids compared to Control (NT). Scale bar, 150 μm ($n = 4$ independent experiments; 30 gastruloids/condition).

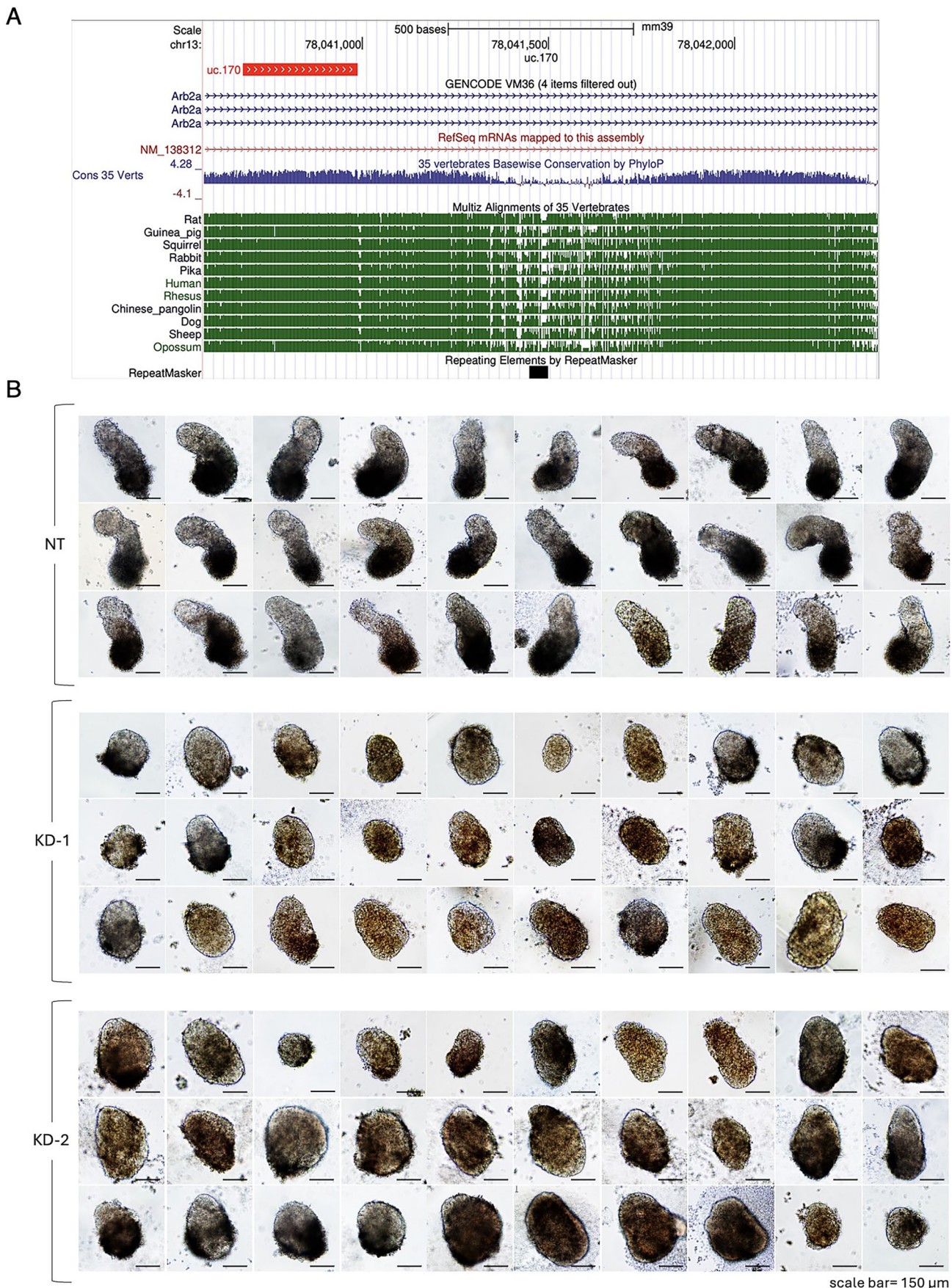

scale bar= 150 μm

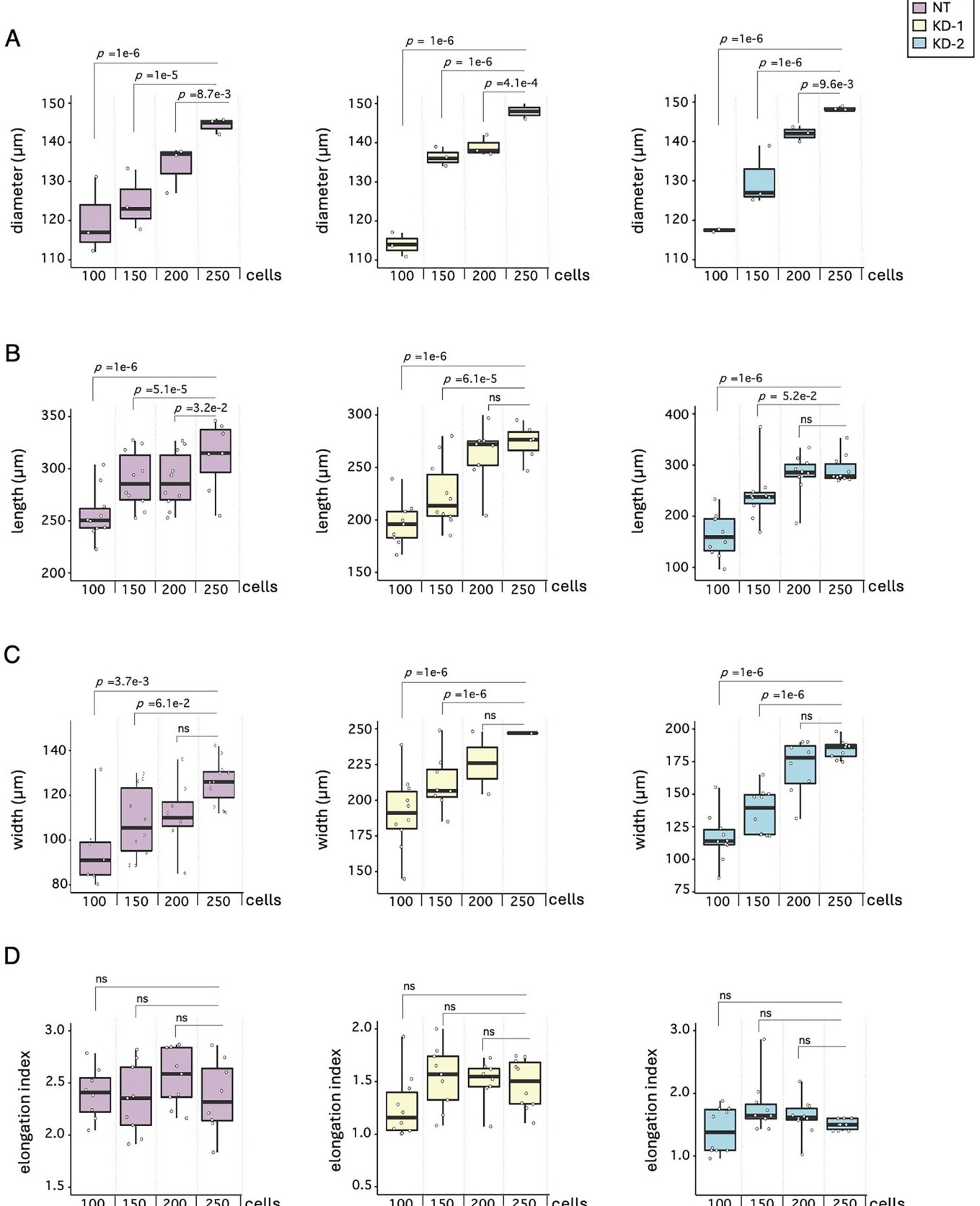

◀ **Figure EV2.** *T-UCstem1* **KD effect is independent on the initial cell number.**

(A) Boxplot diagrams of the aggregate diameter at 48 h of NT (left), KD-1 (middle) and KD-2 (right). Boxplots display the minimum, first quartile, median, third quartile, and maximum. Statistical significance was assessed by one-way ANOVA with Tukey's multiple comparison test. *P* values of ≤0.05 were considered statistically significant. (B–D) Boxplot diagrams of the gastruloids length (B), width (C) and elongation index (D) at 120 h of NT (left), *T-UCstem1* KD-1 (middle) and *T-UCstem1* KD-2 (right). Data are shown as mean ± SD (*n* = 2 independent experiments; 10 gastruloids/condition). Boxplots display the minimum, first quartile, median, third quartile, and maximum. Statistical significance was assessed by one-way ANOVA with Tukey's multiple comparison test. *P* values of ≤0.05 were considered statistically significant.

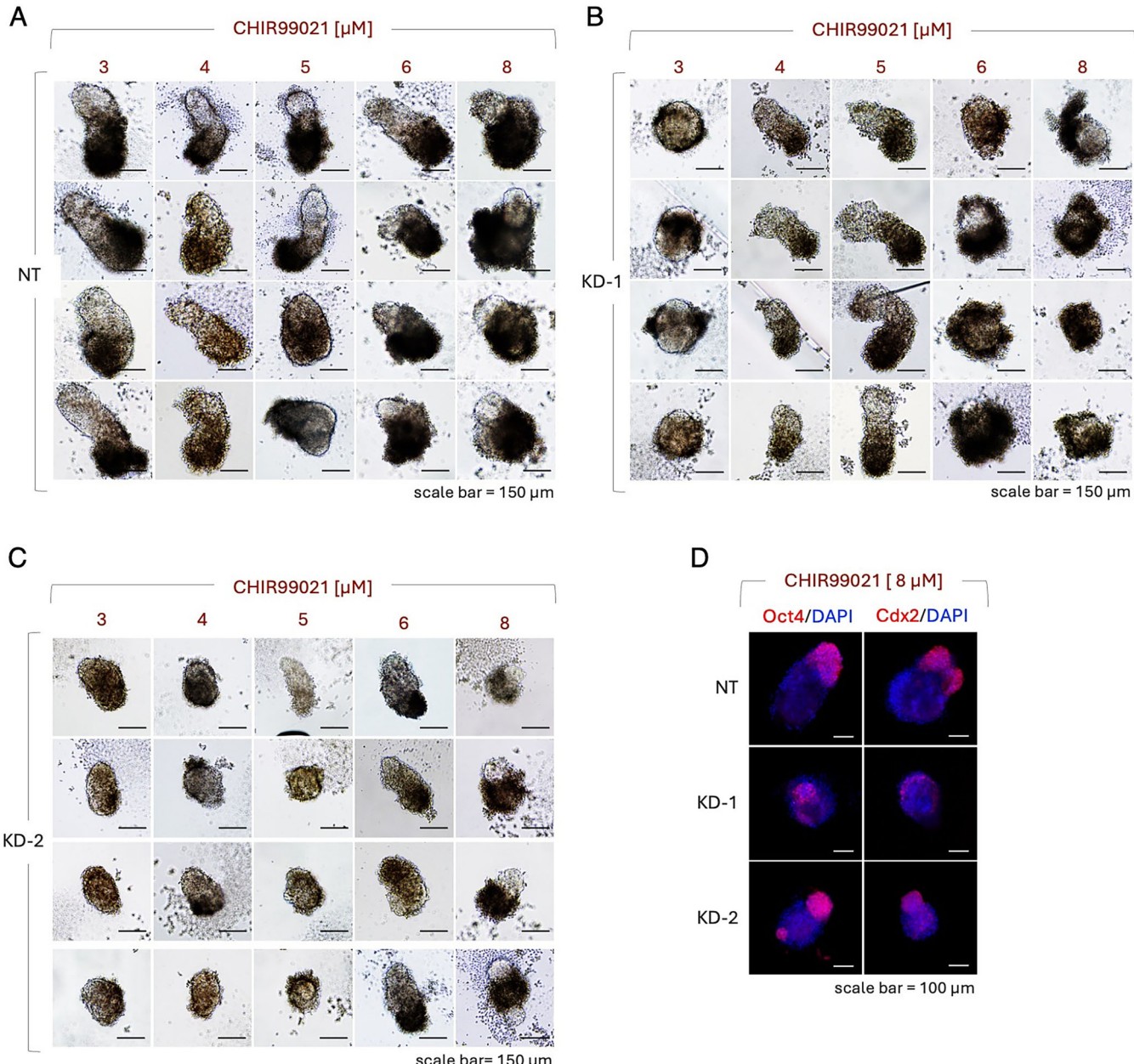

**Figure EV3.  Dose-dependent effect of CHIR99021 on gastruloid development.**

(A–C) Representative brightfield images of Control (NT) (A), KD-1 (B), KD-2 (C) gastruloids treated with increasing concentration of CHIR99021 (3–8 μM). Scale bar, 150 μm. (D) Representative confocal images of Oct4 (pluripotency marker) and Cdx2 (differentiation marker) in gastruloids treated with 8 μM of CHIR99021. Nuclei were counterstained with DAPI (blue). Scale bar, 100 μm. All images are representative of 4 individual gastruloids.

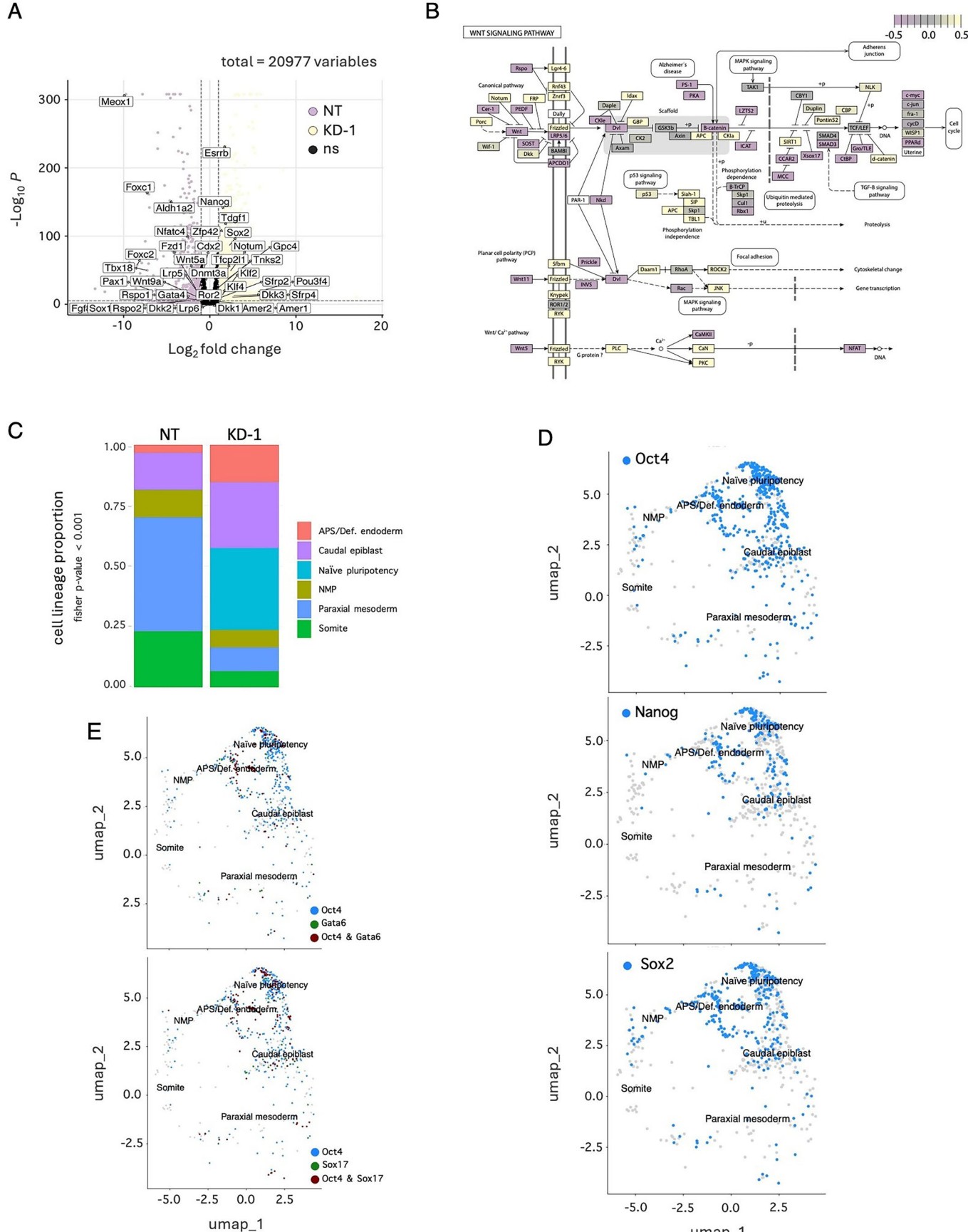

◀ **Figure EV4. Diverse cellular composition in *T-UCstem1* KD gastruloids versus the Control.**

(A) Volcano plot of differentially expressed genes between *T-UCstem1* KD-1 and NT gastruloids. Differential expression analysis was performed using DESeq2 (Wald test; $n = 3$/group). Genes with adjusted $p < 0.05$ and |$\log_2$ fold change| > 1 are color-coded. (B) Reference map of the WNT Signaling pathway from the KEGG pathway database; annotated features are coloured based on differential expression between KD and Control (NT). (C) Stacked barplots of the cell lineage proportion in *T-UCstem1* KD-1 and NT gastruloids. (D) UMAP plot highlighting expression of *Oct4*, *Nanog* and *Sox2* in *T-UCstem1* KD-1 gastruloids. (E) UMAP plot highlighting expression of *Oct4*, *Gata6* and *Sox17* in *T-UCstem1* KD-1 gastruloids.

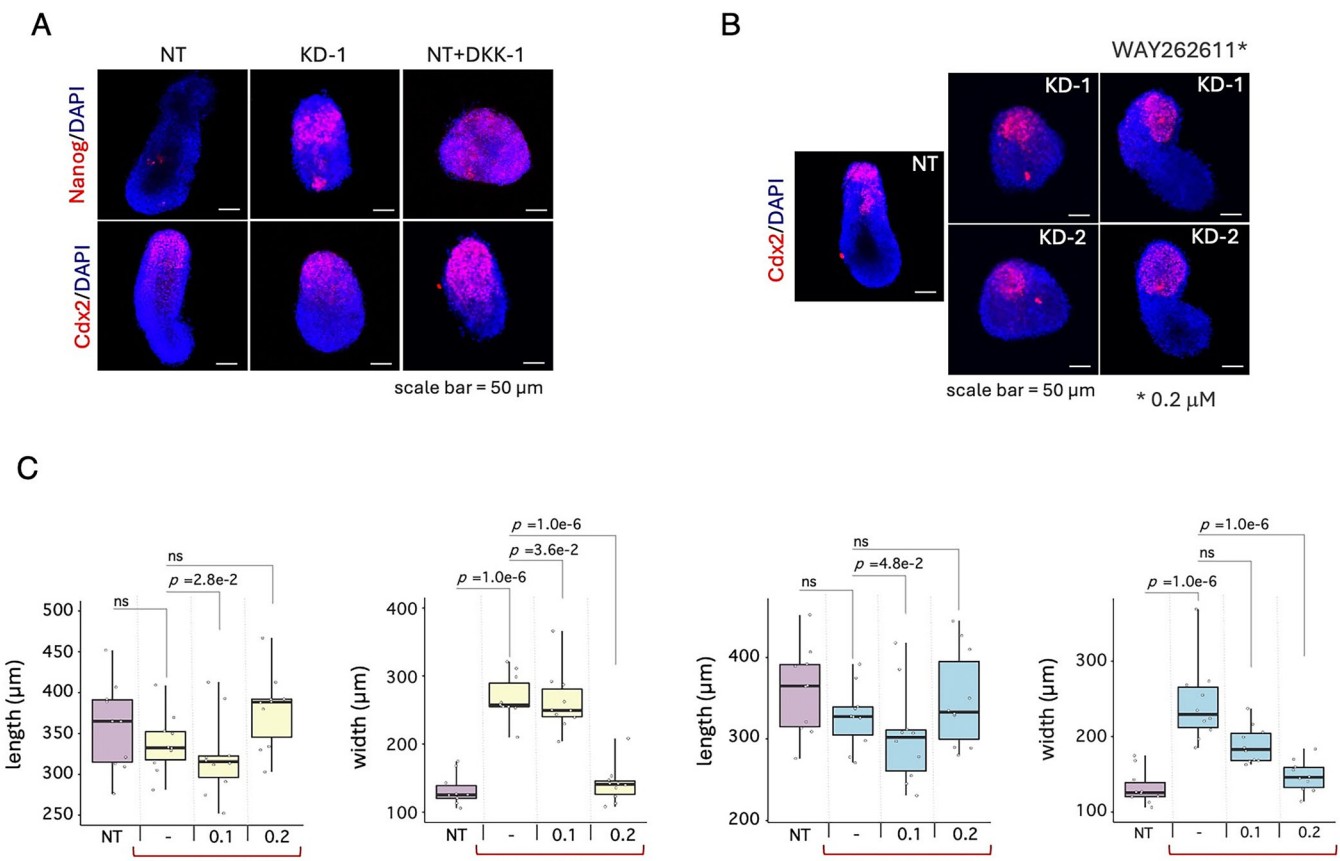

**Figure EV5. DKK-1 blocks the correct gastruloid development.**

(**A**) Representative confocal images of Nanog (pluripotency marker) and Cdx2 (differentiation marker) in gastruloids ± DKK-1 recombinant protein. Nuclei were counterstained with DAPI (blue). Scale bar, 100 μm. All images are representative of 4 individual gastruloids. (**B**) Representative confocal images of Cdx2 (differentiation marker) in gastruloids NT and *T-UCstem1* KD ± WAY262611 at 0.2 μM. Nuclei were counterstained with DAPI (blue). Scale bar, 50 μm. All images are representative of 4 individual gastruloids. (**C**) Boxplot diagrams of gastruloids length (left), width (right) ± WAY262611 at 120 h. Data are shown as mean ± SD ($n = 3$ independent experiments, 10 gastruloids/condition). Boxplots display the minimum, first quartile, median, third quartile, and maximum. Statistical significance was assessed by one-way ANOVA with Tukey's multiple comparison test. *P* values of ≤0.05 were considered statistically significant.

A

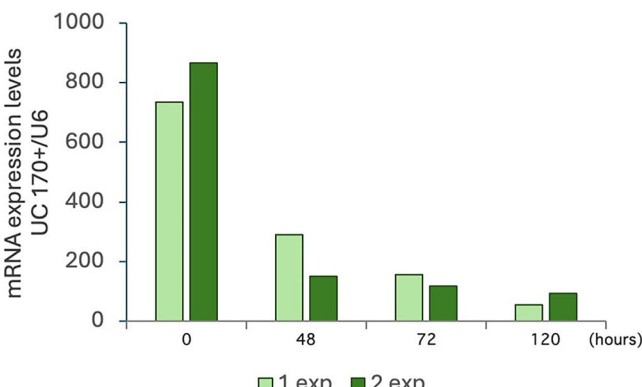
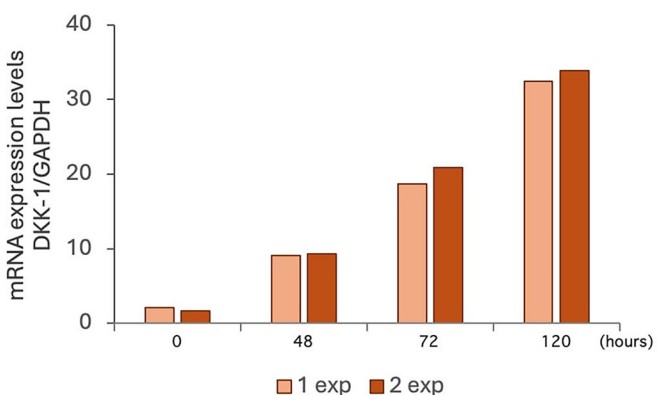

B

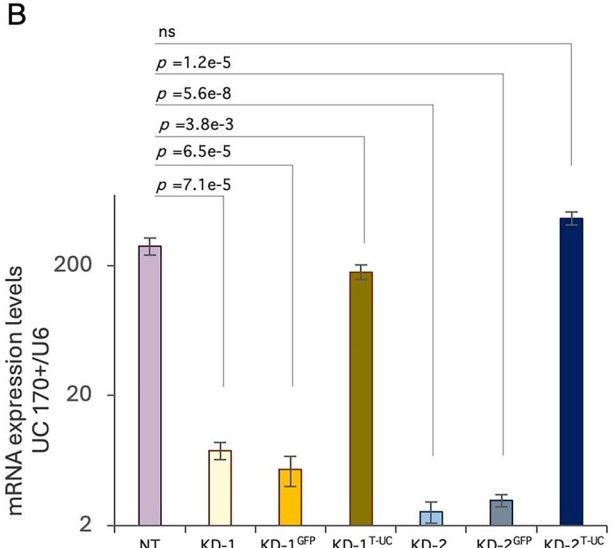
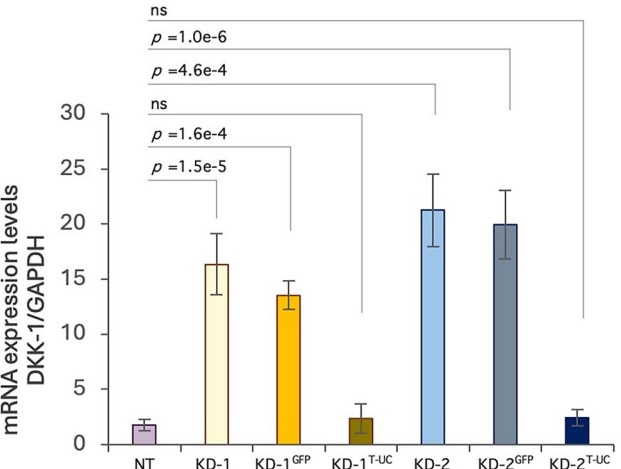

**Figure EV6. Expression analysis of *T-UCstem1* and DKK-1 in mESCs and during gastruloid development.**

(**A**) qRT–PCR analysis of *T-UCstem1* (left) and *DKK-1* (right) expression levels at different stages of gastruloid development in control (NT) conditions. *T-UCstem1* levels were normalized to *U6*, and *DKK-1* levels to *Gapdh*. Individual values of two biological replicates are shown (Exp 1 and 2). (**B**) qRT–PCR analysis of *T-UCstem1* (left) and *DKK-1* (right) expression levels in control (NT), *T-UCstem1* knockdown (KD), KD$^{GFP}$, and KD$^{T-UC}$ mESCs. *T-UCstem1* expression was normalized to *U6*, and *DKK-1* expression to *Gapdh* ($n = 2$ independent experiments). Data are shown as mean ± SD ($n = 3$ independent experiments). Statistical significance was assessed by one-way ANOVA with Tukey's multiple comparison test. *P* values of ≤0.05 were considered statistically significant. Source data are available online for this figure.

