## [Peer Review File · The EMBO Journal]

Non-cell-autonomous control of mouse gastruloid development by the ultra-conserved lncRNA *T-UCstem1*

Annalisa Fico, Arianna Coppola, Filomena Amoroso, Federica Saracino, Gennaro Andolfi, Edoardo Sozzi, Paolo Salerno, Pietro Zoppoli, Alessandro FIORENZANO, Giuseppe Merla, Eduardo Patriarca, and Gabriella Minchiotti

Corresponding author(s): Annalisa Fico (annalisa.fico@igb.cnr.it) , Gabriella Minchiotti (gabriella.minchiotti@igb.cnr.it)

Review Timeline:

Submission Date:	17th Oct 24
Editorial Decision:	21st Jan 25
Revision Received:	2nd May 25
Editorial Decision:	26th Jun 25
Revision Received:	9th Jul 25
Accepted:	13th Aug 25

Editor: Ioannis Papaioannou

Transaction Report:

Dear Annalisa,

Thank you again for submitting your manuscript EMBOJ-2024-119368-T for consideration by The EMBO Journal, and for your patience during peer review. I would like to sincerely apologize once again for the unusually protracted review process in this case, but your manuscript has now been seen by three experts in the field, and we have received the full set of their comments, which are included below.

Referees #1 and #2, whose comments I have already shared with you, are overall supportive of the work mentioning that the data are of good quality and the manuscript interesting and well-prepared. They also identify a number of limitations, however, and identify some aspects that should be further clarified and strengthened by performing additional experimental work. Referee #3, on the other hand, is more critical and -although he/she also indicates interest in the study- points out missing controls in some experiments, as well as insufficient phenotypic characterization and lack of mechanistic understanding.

On balance, and taking into consideration all referees' comments and recommendations, as well as our understanding that the specific comments of the referees are relevant and largely addressable, I would like to invite you to submit a thoroughly revised version of the manuscript along with a detailed point-by-point response addressing all referees' comments. I would kindly request you take all comments of the referees on board during revising your manuscript. Although we agree with referee #3 that some further mechanistic insight would be desirable and increase the impact of your work on the field, no new far-reaching experiments for a complete mechanistic understanding will be required for further consideration of the study at The EMBO Journal. However, all technical issues and points relevant to the solidity of the conclusions and model should be sufficiently addressed.

I should add that it is The EMBO Journal policy to allow only a single round of major revision, and acceptance of your manuscript will therefore depend on the completeness of your responses in this revised version. Please let me know if you would like to discuss with me further any particular points from the referees' reports, or if you have any questions or other comments.

We generally allow three months as standard revision time (April 20, 2025). Should you foresee a problem in meeting this three-month deadline, please let us know in advance and we may be able to grant an extension. As a matter of policy, competing manuscripts published during this period will not negatively impact our assessment of the conceptual advance presented by your study. However, we request that you contact us as soon as possible upon publication of any related work, to discuss how to proceed.

Thank you for the opportunity to consider your work for publication in The EMBO Journal. I look forward to your revision.

Best wishes,

Ioannis

Instructions for preparing your revised manuscript

1. When you are ready to submit the revision, please upload:

- A Word file of the manuscript text (including legends of main Figures, EV Figures and Tables). Please make sure that changes are highlighted (or "tracked") to be clearly visible.

- Individual production-quality figure files (one file per figure). When assembling your figures, please refer to our figure preparation guidelines in order to ensure proper formatting and readability in print as well as on screen:

If the data shown in a figure are obtained from n {less than or equal to} 2, please use scatter plots showing the individual data points.

i. the name of the statistical test used to generate error bars and P values

ii. the number (n) of independent experiments (please specify technical or biological replicates) underlying each data point (discussion of statistical methodology can be reported in the Materials and Methods section, but figure legends should contain a basic description of n, P, and the test applied)

iii. the nature of the bars and error bars (s.d., s.e.m.).

- A point-by-point response to the referees' comments, with a detailed description of the changes made (as a word file). All referees' concerns must be fully addressed and their suggestions taken on board. When preparing your letter of response to the referees' comments, please bear in mind that this will form part of the Review Process File and will therefore be available online to the community. Please note that you have the possibility to opt out of the transparent process at any stage prior to publication by letting the editorial office know (contact@embojournal.org); if you do opt out, the Review Process File link will point to the following statement: "No Review Process File is available with this article, as the authors have chosen not to make the review process public in this case.". For more details on our Transparent Editorial Process, please visit our website: <https://www.embopress.org/page/journal/14602075/authorguide#transparentprocess>

- Expanded View (EV) files (replacing Supplementary Information) that are collapsible/expandable online. A maximum of 5 EV Figures can be typeset. EV Figures should be cited as "Figure EV1, Figure EV2" etc. in the text, and their respective legends should be included in the manuscript file after the legends of regular figures. See detailed instructions regarding Expanded View files here:

- For the figures that you do NOT wish to display as Expanded View figures, they should be bundled together with their legends in a single PDF file called "Appendix", which should start with a short Table of Contents (including page numbers). Appendix figures should be referred to in the main text as: "Appendix Figure S1, Appendix Figure S2" etc. Please see detailed instructions here: <https://www.embopress.org/page/journal/14602075/authorguide#expandedview>

- A complete author checklist, which you can download from our author guidelines (<https://www.embopress.org/page/journal/14602075/authorguide>). Please note that the checklist will also be part of the Review Process File.

2. Please note that no statistics should be calculated and shown in Figures if n=2. Please also note that each p value should be reported as an exact value.

3. Before submitting your revision, primary datasets (and computer code, where appropriate) produced in this study need to be deposited in appropriate public databases (see <https://www.embopress.org/page/journal/14602075/authorguide#dataavailability>).

In particular, we kindly request you to deposit all RNA sequencing data produced in this study in appropriate databases. If new code was generated for data analysis, it should also be made publicly available. The accession numbers, database, and the specific URLs (links) should be listed in a formal "Data availability" section (placed after Methods), following the example below:

"The RNA-seq datasets produced in this study are available in the following database:

Gene Expression Omnibus GSE46843 (<https://www.ncbi.nlm.nih.gov/geo/query/acc.cgi?acc=GSE46843>)"

*** All links should resolve to a page where the data can be accessed. ***

*** Please remember to provide in the Data availability section of your revised manuscript reviewer passwords if the datasets are not yet public. ***

*** The Data Availability Section is restricted to new primary data that are part of this study. In case you have no data that require deposition in a public database, please state so instead of referring to the database: "Our study includes no data deposited in public repositories." under the heading "Data availability". ***

4. Please check that the title and the abstract of the manuscript are brief, yet explicit, even to non-specialists. The length of the title should not exceed 100 characters, and the abstract should be a single paragraph not exceeding 175 words.

5. Please also note our reference format: <https://www.embopress.org/page/journal/14602075/authorguide#referencesformat>.

7. Please remember: digital image enhancement is acceptable practice, as long as it accurately represents the original data and conforms to community standards. If a figure has been subjected to significant electronic manipulation, this must be noted in the figure legend or in the "Materials and Methods" section. The editors reserve the right to request original versions of figures and

the original images that were used to assemble the figure.

8. Our journal encourages inclusion of data citations in the reference list to directly cite datasets that were obtained from public databases. Data citations in the article text are distinct from normal bibliographical citations and should directly link to the database records from which the data can be accessed. In the main text, data citations are formatted as follows: "Data ref: Smith et al, 2001" or "Data ref: NCBI Sequence Read Archive PRJNA342805, 2017". In the Reference list, data citations must be labeled with "[DATASET]". A data reference must provide the database name, accession number/identifiers, and a resolvable link to the landing page from which the data can be accessed at the end of the reference. Further instructions are available at: <https://www.embopress.org/page/journal/14602075/authorguide#referencesformat>.

9. We request authors to consider both actual and perceived competing interests. Please review our policy (<https://www.embopress.org/page/journal/14602075/authorguide#conflictsofinterest>) and update your competing interests statement if necessary. Please name this section 'Disclosure and competing interests statement' and place it after the Acknowledgements section.

10. Please note that all corresponding authors are required to provide an ORCID ID upon submission of a revised manuscript (<https://orcid.org/>). Please find instructions on how to link your ORCID ID to your account in our manuscript tracking system in our Author guidelines (<https://www.embopress.org/page/journal/14602075/authorguide#authorshipguidelines>).

11. We use CRediT to specify the contributions of each author in the journal submission system. CRediT replaces the author contribution section, which should be removed from the manuscript. Please use the free text box to provide more detailed descriptions. See also guide to authors: <https://www.embopress.org/page/journal/14602075/authorguide#authorshipguidelines>.

13. We would also welcome the submission of cover suggestions or motifs to be used by our Graphics Illustrator in designing a cover.

14. Please use the link below to submit your revision:
<https://emboj.msubmit.net/cgi-bin/main.plex>

Referee #1:

In this manuscript, Coppola et al. investigated the role of the lncRNA T-UCstem1 during gastruloids development. This manuscript follows up on a previous study from the laboratory, where T-UCstem1 had been identified as a regulator of Embryonic Stem Cell (ESCs) self-renewal and transcriptional identity. Here the authors investigate the function of T-UCstem1 during the differentiation and self-organization of ESC into gastruloids, a model for post-implantation mammalian embryos. They identify that T-UCstem1 is required for gastruloids development as, in knocked-down experiments and gastruloids formation, pluripotency is maintained and failure to elongate was observed. Then they provide evidence that this effect is mediated by DKK1, a WNT-inhibitor and that the phenotype can be circumvented by using higher concentration of the WNT agonist (CHIR99021), a DKK1 inhibitor and that control gastruloids fail to develop in presence of exogenously provided DKK1. Altogether the authors provide an interesting and well-written manuscript. The data generated is of good quality and proper quantifications were performed to document their observations. The link between the phenotype and DKK1 expression is clearly demonstrated. However, I have some comments which would need to be addressed before proceeding with publication.

Major comment

1. Throughout the manuscript, the authors do not assess the expression of T-UCstem1 during gastruloids development. Is T-UCstem1 expressed during gastruloids development or is this effect caused from differences at the ESC status? Since ESC clone isolations can often lead to clonal specific defect, ideally, in order to confirm the implication of T-UCstem1 in the up-regulation of DKK1, a rescue experiment could be done to confirm the implication of T-UCstem1 in gastruloids development. For example, knocking out the promoter of the shRNA in one of the clones, or by over-expressing T-UCstem1 in one of the clone. I understand this is a major experiment, but in order to confirm that the phenotype is indeed driven by the loss of T-UCstem1 transcript, it would be crucial to connect the phenotype to the loss of this lncRNA.

2. Each clone appears to require a specific CHIR concentration, 4 and 5uM being optimal for clone 1 and 6uM for clone two. Do these reflect different knock-down efficiency of T-UCstem1 for each clone. Are there any explanation as to why at 6uM clone 1 isn't rescued anymore? Since NT gastruloids are only weakly impacted by the use of 6uM CHIR, this shouldn't be in the toxic range yet.

3. The annotation of the single cell RNAseq is unexpected. The presence of ExE ectoderm, Exe Endoderm, Exe Mesoderm is quite surprising as gastruloids being generated from ESC, they shouldn't have the ability to differentiate into extra-embryonic tissue. Furthermore, to my knowledge, this is not cell identities which have been previously observed in other single cell datasets from gastruloids. What does NMP embryo means? The presence of late erythroid cells (Erythroid 3) is surprising as well. Generally, the authors should provide much more details about how their clusters were annotated and of their single cell analysis.

4. Maybe I am misunderstanding something but in Fiorezano et al. 2018, my understanding is that T-UCstem1 KD cells grown in presence of FBS Lif and feeder are unable to grow into dome shape colonies and differentiation (to neurons) is promoted in KD ESC. In contrast, in this study where cells are grown in FBS Lif 2i, the opposite is observed. KD cells can make domed colonies as efficiently as the non-treated cells, and in contrast to the previous study, differentiation into gastruloids is hindered and pluripotency is promoted. Could the author provide some explanations to this divergence between these conditions yielding an opposite effect?

5. Regarding the double staining analysis of dissociated gastruloids in Fig. 6D. Was there any reason to analyse this co-expression in dissociated gastruloids? Why wasn't this done in wholemount as was performed like in the rest of the manuscript?

Minor comment

Abstract l28 knock-down acronym should be defined there since it is used l30

Figure 5C: the text y-axis is not useful it should be removed.

Referee #2:

The manuscript by Copolla explore the role of ultraconserved long non-codingRNAs in early mammalian embryogenesis. The authors explored the lncRNA named T-UCstem1.

The results are clearly presented and the figures are well structured.

The mechanism identified is new.

This paper is of interest for the readers of the journal

Some comments:

1 the way the authors selected this T-UC is not explained. This should be the starting paragraph of results.

2 a diagram with the conservation of this ultraconserved sequence among species is helpful

3 the first results paragraph is performed with knockdown experiments. Did the author try to restore the expression of the T-UCstem1 in the KD cells and find out what are the changes in the phenotype?

4 it is a regulation feedback between T-UCstem1 and DKK-1 expression? Did these two molecules bind directly?

5 what is the expression of T-UCstem1 in other stages of embryonic development?

Referee #3:

In this manuscript, Coppola et al explore the role of the lncRNA T-UCstem1 during mouse gastrulation, using gastruloids as a model system. The role of lncRNAs during early mouse embryo development is poorly understood, and therefore this is a question of interest for the field. Their findings indicate that T-UCstem1 is needed for gastruloid elongation and proper cell fate specification. While interesting, the study is still quite preliminary. Appropriate controls and a careful detailed analysis of the phenotype would be needed. Moreover, in the present form, the article lacks a mechanistic understanding of T-UCstem1 function. For these reasons, I do not recommend publication in the EMBO journal. My specific comments are the following:

Major comments:

1. The study begins by knocking down T-UCstem1 in mouse ESCs. No validation of the KD is provided. This is particularly important, as clones 1 and 2 have slightly different phenotypes. What are the precise levels of T-UCstem1 in ESCs and gastruloids? How long does the KD last? Do T-UCstem1 levels increase by 96-120 hours of the gastruloid protocol? The authors would also need to rescue the phenotype by re-introducing T-UCstem1. The starting KD ESCs are also not characterized (beyond a colony formation assay). Are there gene expression changes already in the naïve state or do these only manifest upon naïve pluripotency exit? What is the rate of proliferation of control and KD cells in naïve conditions and gastruloids? This is important as the authors explore differences in numbers as a potential cause of the phenotype. Globally, these are all critical controls that are currently missing.

2. Reproducibility: throughout the article, the number of gastruloids analyzed and the number of independent experiments are

not mentioned. The IF images are not analyzed. Therefore, the reproducibility of the images shown and/or variability of the phenotype is not clear. In the CHIR rescue only OCT4 and CDX2 expression are shown. A complete analysis of markers and single cell quantification would be needed to support the rescue experiment. Moreover, the methods section lacks enough detail (e.g., media recipes, catalog numbers, etc).

3. The model the authors propose is that increased expression of DKK1 in T-UCstem1 KD cells is responsible for the defects observed in the gastruloids. What are the levels of Wnt activity? Using a Wnt reporter ESC line would be informative. This is important as Brachyury is expressed in the KD EBs, which suggests Wnt signaling is active. Moreover, the increase in DKK1 expression shown in Figure 7D seems to be small in clone 1. What are the protein levels of DKK1? The DKK1 experiments (Figure 8) should also show a detailed molecular characterization.

4. Defects in pluripotency exit: from the data presented is not clear whether loss of T-UCstem1 causes a naïve pluripotency exit defect or a defect in lineage specification. The colony formation assay in 2iLIF tests for naïve pluripotency, while the IF data presented by the authors analyses Oct4, which is a core pluripotency marker. What is exactly happening in terms of pluripotency progression? Is naïve pluripotency exit compromised? Are there cells expressing naïve pluripotency markers in KD gastruloids? Are formative pluripotency markers expressed? The pluripotent status of the cells should be clearly defined.

5. Mechanism: how does T-UCstem1 control DKK1 expression? Some level of mechanistic insight should be expected for publication in the EMBO journal.

Minor comments:

1. The manuscript needs careful editing.

2. The limitations of gastruloids should also be acknowledged in the introduction.

3. Figure 5F: similar plots should be shown for the control gastruloids.

4. Figure 6D: the use of the cytopsin is not ideal. If possible, the authors should image and quantify specific planes within the gastruloids. The study would also benefit from higher magnification and resolution images.

5. The authors conclude that the KD gastruloids display a wide range of shapes and sizes but this is true only for clone 2, not clone 1.

6. Line 446: The elevated levels of DKK1 found in patients with unexplained miscarriages are seen in the endometrium, not the embryo. This statement is misleading.

EMBOJ-2024-119368-T

Response to Reviewer #1:

In this manuscript, Coppola et al. investigated the role of the lncRNA *T-UCstem1* during gastruloids development. This manuscript follows up on a previous study from the laboratory, where *T-UCstem1* had been identified as a regulator of Embryonic Stem Cell (ESCs) self-renewal and transcriptional identity. Here the authors investigate the function of *T-UCstem1* during the differentiation and self-organization of ESC into gastruloids, a model for post-implantation mammalian embryos. They identify that *T-UCstem1* is required for gastruloids development as, in knocked-down experiments and gastruloids formation, pluripotency is maintained and failure to elongate was observed. Then they provide evidence that this effect is mediated by DKK1, a WNT-inhibitor and that the phenotype can be circumvented by using higher concentration of the WNT agonist (CHIR99021), a DKK1 inhibitor and that control gastruloids fail to develop in presence of exogenously provided DKK1. Altogether the authors provide an interesting and well-written manuscript. The data generated is of good quality and proper quantifications were performed to document their observations. The link between the phenotype and DKK1 expression is clearly demonstrated. However, I have some comments which would need to be addressed before proceeding with publication.

We thank the Reviewer for his/her positive overall comment on the present work.

Major comment

1. Throughout the manuscript, the authors do not assess the expression of *T-UCstem1* during gastruloids development. Is *T-UCstem1* expressed during gastruloids development or is this effect caused from differences at the ESC status?

We thank the Reviewer for this insightful comment. We have analyzed the expression levels of *T-UCstem1* by real-time PCR during gastruloid development. The results indicate that *T-UCstem1* is expressed during gastruloid development and that its expression levels progressively decrease over time.

Interestingly, we observed that the expression profile of *T-UCstem1* inversely correlates with that of DKK-1, supporting our proposed model by which *T-UCstem1* negatively regulates DKK-1 expression and thereby modulates the WNT signaling pathway.

These results are now shown in Figure EV6A of the revised manuscript and reported on page 13, lines 292–294.

EMBOJ-2024-119368-T

Since ESC clone isolations can often lead to clonal specific defect, ideally, in order to confirm the implication of T-UCstem1 in the up-regulation of DKK1, a rescue experiment could be done to confirm the implication of T-UCstem1 in gastruloids development. For example, knocking out the promoter of the shRNA in one of the clones, or by over-expressing T-UCStem1 in one of the clone. I understand this is a major experiment, but in order to confirm that the phenotype is indeed driven by the loss of T-UCstem1 transcript, it would be crucial to connect the phenotype to the loss of this lncRNA.

We thank the Reviewer for this valuable suggestion.

To address this comment, we performed a rescue experiment by overexpressing the ultraconserved element uc.170+ in both *T-UCstem1* knockdown ESC clones (KD1 and KD2). Specifically, a bicistronic expression plasmid encoding both uc.170+ and GFP under the control of the ubiquitous CAG promoter was used (Pascale E, et al. *Stem Cell Reports*. 2020;15(4):836–844. doi:10.1016/j.stemcr.2020.08.009). A plasmid expressing only GFP served as a control. Using these constructs, we generated four new ESC lines: KD-1^{GFP}, KD-1^{T-UC}, KD-2^{GFP}, and KD-2^{T-UC}.

Notably, re-expression of the lncRNA in both KD1 and KD2 ESC lines restored DKK-1 expression to wild-type levels and fully rescued the ability of KD1 and KD2 ESCs to generate elongated gastruloids in the presence of 3 μ M CHIR.

These Results are now shown in Figure 9 and EV6B and reported on page 13, lines 294-298 of the revised manuscript.

We appreciate the Reviewer's thoughtful comment, which helped strengthen the conclusions of our study.

2. Each clone appears to require a specific CHIR concentration, 4 and 5 μ M being optimal for clone 1 and 6 μ M for clone two. Do these reflect different knock-down efficiency of T-UCstem1 for each clone. Are there any explanation as to why at 6 μ M clone 1 isn't rescued anymore? Since NT gastruloids are only weakly impacted by the use of 6 μ M CHIR, this shouldn't be in the toxic range yet.

We thank the Reviewer for this observation.

The different CHIR concentration required to rescue gastruloid elongation in each KD line likely reflect differences in *T-UCstem1* knockdown efficiency (Fiorenzano et al., 2018) as well as a clear difference in DKK-1 expression levels (see revised manuscript, Figure 7D and EV6B). Specifically, KD-2 exhibits higher DKK-1 expression compared to KD-1, which likely necessitates a higher CHIR concentration (6 μ M) to counterbalance the increased WNT inhibition and effectively rescue the phenotype.

Conversely, in KD-1, 6 μ M CHIR likely exceeds the optimal threshold of WNT activation needed for proper gastruloid elongation, which is instead achieved with 5

EMBOJ-2024-119368-T

μM CHIR in this KD clone. These results underscore the critical importance of finely tuned WNT signaling in developmental patterning.

Moreover, as shown in EV Figure 3, higher concentrations of CHIR (ranging from 5 μM up to 8 μM) induce partial toxicity in NT gastruloids, as evidenced by an increase in dead cells surrounding the anterior pole of the aggregates.

3. The annotation of the single cell RNAseq is unexpected. The presence of ExE ectoderm, Exe Endoderm, Exe Mesoderm is quite surprising as gastruloids being generated from ESC, they shouldn't have the ability to differentiate into extra-embryonic tissue. Furthermore, to my knowledge, this is not cell identities which have been previously observed in other single cell datasets from gastruloids.

We thank the Reviewer for raising this important point. We were also surprised to find ExE (Ectoderm, Endoderm, and Mesoderm) populations in the SC RNA-Seq. To achieve a comprehensive annotation, we integrated the mouse embryo atlas from Pijuan-Sala et al., 2019 (DOI: 10.1038/s41586-019-0933-9) with the gastruloid atlas from Suppinger et al., 2023 (DOI: 10.1016/j.stem.2023.04.018), as detailed in the Methods section. Although ExE-related populations originate from the embryo-based annotation, we opted to retain these labels for a more exhaustive and unbiased analysis. In this context, a study published during the revision process also reported the presence of ExE populations in gastruloids (Argiro et al., Nature Comm. 2024, DOI: 10.1038/s41467-024-54466-w). They do note, however, that for their own analysis “*an atlas must be comparable to the dataset under study*” and therefore excluded “endoderm” and “extraembryonic endoderm” populations. While we acknowledge this approach, we chose to maintain a broader annotation framework to allow for a more complete and explorative analysis—especially given the potential changes in cell identities under knockdown conditions.

What does NMP embryo means? The presence of late erythroid cells (Erythroid 3) is surprising as well. Generally, the authors should provide much more details about how their clusters were annotated and of their single cell analysis.

We thank the Reviewer for pointing out the potential confusion. In our dataset, we used two annotations for NMPs: one from Pijuan-Sala et al., 2019 and one from Suppinger et al., 2023. To clarify this for the reader, we have renamed these clusters to “NMP” and “NMP2” respectively, and have updated the text accordingly (see revised manuscript, page 9, lines 198-201, and Figure 5D-F and EV4C, D).

As for the population labeled as late erythroid cells (Erythroid 3), the same rationale applies: although these labels stem from embryo-based annotations, we retained them to preserve analytical completeness and reduce bias.

EMBOJ-2024-119368-T

Finally, in response to the request for more detail, we have significantly expanded the Methods section to include more detailed information on how cluster annotation was performed (see revised manuscript, page 23, lines 515-525) including also a supplementary file (*SC-RNA-seq annotation scores*) with the annotation scores of each signature label for each cluster. This file will allow an interested reader to self-evaluate the strength of the label call and the possible alternatives.

4. Maybe I am misunderstanding something but in Fiorenzano et al. 2018, my understanding is that T-UCstem1 KD cells grown in presence of FBS Lif and feeder are unable to grow into dome shape colonies and differentiation (to neurons) is promoted in KD ESC. In contrast, in this study where cells are grown in FBS Lif 2i, the opposite is observed. KD cells can make domed colonies as efficiently as the non-treated cells, and in contrast to the previous study, differentiation into gastruloids is hindered and pluripotency is promoted. Could the author provide some explanations to this divergence between these conditions yielding an opposite effect?

We thank the Reviewer for the opportunity to clarify this important point. The Reviewer is correct that under standard FBS/LIF conditions and in the presence of feeders, *T-UCstem1* KD ESC cells generate flat colonies and enhanced neuronal differentiation, as reported in Fiorenzano et al. 2018. The apparent discrepancy between these findings and those presented in this study is explained by the distinct culture conditions used. Indeed, as already reported in Fiorenzano et al., 2018, the addition of *2i* to the culture medium was sufficient to rescue the dome-shaped colony morphology of KD cells that also stained positive for alkaline phosphatase (see Figure 4E of Fiorenzano et al.). This morphological rescue is further supported by molecular data indicating restored expression of key pluripotency markers, such as *Nanog* and *Sox2* (Figure 4F in Fiorenzano et al., 2018).

Therefore, the observations reported in the current study are fully consistent with the previous findings, as the colony formation assay—which precedes the gastruloid formation assay—is performed under *2i* conditions.

5. Regarding the double staining analysis of dissociated gastruloids in Fig. 6D. Was there any reason to analyse this co-expression in dissociated gastruloids? Why wasn't this done in wholemout as was performed like in the rest of the manuscript?

We thank the Reviewer for this question. We chose to perform the double staining analysis on dissociated gastruloids in order to allow for a more precise and unambiguous assessment of *Oct4/Brachyury* co-expression at the single-cell level. While wholemout imaging provides valuable spatial context, it can sometimes make it difficult to definitively determine whether two markers are co-expressed within the

EMBOJ-2024-119368-T

same cell due to overlapping signals in densely packed regions. By dissociating the gastruloids, we ensured a clearer resolution and more confident identification of double-positive cells. This approach complemented the wholemount analyses presented in the manuscript and strengthened our interpretation of co-expression dynamics.

Minor comment

Abstract I28 knock-down acronym should be defined there since it is used I30

We have now defined the knock-down acronym.

Figure 5C: the text y-axis is not useful it should be removed.

We have removed the unnecessary text from the y-axis.

Response to Reviewer #2:

The manuscript by Copolla explore the role of ultraconserved long non-codingRNAs in early mammalian embryogenesis. The authors explored the lncRNA named T-UCstem1.

The results are clearly presented and the figures are well structured.

The mechanism identified is new.

This paper is of interest for the readers of the journal

We thank the Reviewer for the positive overall comment on the present work.

Some comments:

1. the way the authors selected this T-UC is not explained. This should be the starting paragraph of results.

We thank the Reviewer for this comment. The rationale for selecting *T-UCstem1* was indeed a key focus of a previous study (Fiorenzano A, et al. Stem Cell Reports. 2018 Mar 13;10(3):1102-1114. doi: 10.1016/j.stemcr.2018.01.014), in which we reported the screening strategy among the transcribed ultraconserved elements, based on their expression patterns in ESC differentiation.

This has been now clearly reported in the revised manuscript on page 5, lines 91–95.

2. a diagram with the conservation of this ultraconserved sequence among species is helpful

We thank the Reviewer for this helpful suggestion. Following this recommendation, we analysed the conservation of the entire lncRNA containing the ultraconserved element uc.170 across multiple species. Our analysis reveals that, similar to uc.170, the majority of *T-UCstem1* sequence is highly conserved among vertebrates. We have now included a comparative conservation diagram in the revised Figure EV1A

EMBOJ-2024-119368-T

to illustrate this point and report these findings on page 5, lines 91–95 of the revised manuscript.

Briefly, conservation of the 35 Vertebrates is calculated by phyloP algorithm (Siepel A et al., 2005; PMID: 16024819), which can measure acceleration (faster evolution than expected under neutral drift) as well as conservation (slower than expected evolution). In the phyloP plots, sites predicted to be conserved are assigned positive scores (and shown in blue), while sites predicted to be fast evolving are assigned negative scores (and shown in red) (Figure EV1A). Pairwise alignments of some species to the mouse genome are displayed below the conservation histogram as a wiggle that indicates alignment quality. Note that excluding species from the pairwise display does not alter the conservation score display.

3. the first results paragraph is performed with knockdown experiments. Did the author try to restore the expression of the T-UCstem1 in the KD cells and find out what are the changes in the phenotype?

We thank the Reviewer for this insightful comment.

To address this comment, we performed a rescue experiment by overexpressing the ultra-conserved element uc.170+ in both *T-UCstem1* knockdown ESC clones (KD1 and KD2). Specifically, we used a bicistronic expression plasmid encoding both uc.170+ and GFP under the control of the ubiquitous CAG promoter that has been previously described (Pascale E, et al. *Stem Cell Reports*. 2020;15(4):836–844. doi:10.1016/j.stemcr.2020.08.009). A plasmid expressing only GFP served as a control. Using these constructs, we generated four new ESC lines: KD-1^{GFP}, KD-1^{T-UC}, KD-2^{GFP}, and KD-2^{T-UC}.

Notably, re-expression of the lncRNA in both KD1 and KD2 ESC lines restored DKK-1 expression to wild-type levels and fully rescued the ability of KD1 and KD2 ESCs to generate elongated gastruloids in the presence of 3 μM CHIR.

These Results are now shown in Figure 9 and reported on page 13, lines 294-298 of the revised manuscript.

We greatly appreciate the Reviewer's suggestion, which has helped strengthen the conclusions of our study.

4. it is a regulation feedback between T-UCstem1 and DKK-1 expression? Did these two molecules bind directly?

We thank the Reviewer for raising this question. Several evidence support the existence of a regulatory feedback mechanism between *T-UCstem1* and DKK-1 expression : (i) the inverse correlation in expression patterns of *T-UCstem1* and DKK-1 during gastruloid development (Figure EV6A); (ii) the upregulation of DKK-1 observed in *T-UCstem1* knockdown conditions, both in undifferentiated ESCs and

EMBOJ-2024-119368-T

during gastruloid formation (Figure 7D); and (iii) the restoration of DKK-1 expression levels following *T-UCstem1* overexpression in knockdown cells (Figure EV6B).

We have previously reported that *T-UCstem1* interacts with and stabilizes the PRC2 complex on bivalent chromatin domains to modulate gene expression in undifferentiated ESCs (Fiorenzano A, et al. Stem Cell Reports. 2018 Mar 13;10(3):1102-1114. doi: 10.1016/j.stemcr.2018.01.014). We thus hypothesize that *T-UCstem1* may regulate DKK-1 expression through a similar PRC2-dependent mechanism. However, we cannot rule out the possibility that additional regulatory mechanisms may be involved, including a direct interaction between *T-UCstem1* and the DKK-1 locus or transcript. Further studies are needed to investigate these possibilities in depth, which will undoubtedly be of significant interest.

5. what is the expression of T-UCstem1 in other stages of embryonic development?

We currently do not have data on the expression of *T-UCstem1* in embryonic development. We agree that this is a highly interesting and relevant question, which would necessitate a dedicated study to fully address. However, we would like to mention that uc.170, the ultraconserved element within *T-UCstem1*, acts as an enhancer in the cerebellar regions of E11.5 embryos (Pennacchio L. A. et al., *Nature* 2006, DOI: 10.1038/nature05295), suggesting a potentially crucial role of uc.170 in gene regulation during embryonic development.

Response to Reviewer #3:

In this manuscript, Coppola et al explore the role of the lncRNA T-UCstem1 during mouse gastrulation, using gastruloids as a model system. The role of lncRNAs during early mouse embryo development is poorly understood, and therefore this is a question of interest for the field. Their findings indicate that T-UCstem1 is needed for gastruloid elongation and proper cell fate specification. While interesting, the study is still quite preliminary. Appropriate controls and a careful detailed analysis of the phenotype would be needed. Moreover, in the present form, the article lacks a mechanistic understanding of T-UCstem1 function. For these reasons, I do not recommend publication in the EMBO journal. My specific comments are the following:

Major comments:

1. The study begins by knocking down T-UCstem1 in mouse ESCs. No validation of the KD is provided. This is particularly important, as clones 1 and 2 have slightly different phenotypes. What are the precise levels of T-UCstem1 in ESCs and gastruloids? How long does the KD last? Do T-UCstem1 levels increase by 96-120 hours of the gastruloid protocol? The authors would also

EMBOJ-2024-119368-T

need to rescue the phenotype by re-introducing T-UCstem1. The starting KD ESCs are also not characterized (beyond a colony formation assay). Are there gene expression changes already in the naïve state or do these only manifest upon naïve pluripotency exit? What is the rate of proliferation of control and KD cells in naïve conditions and gastruloids? This is important as the authors explore differences in numbers as a potential cause of the phenotype. Globally, these are all critical controls that are currently missing.

We respectfully disagree with the Reviewer, and apologies for the lack of clarity that may have contributed to the confusion. Most of the concerns regarding the validation and characterization of the KD clones have already been addressed in the manuscript by Fiorenzano et al., 2018 (Fiorenzano A, et al. Stem Cell Reports. 2018 Mar 13;10(3):1102-1114. doi: 10.1016/j.stemcr.2018.01.014), which includes comprehensive controls and detailed molecular and cellular characterization of the *T-UCstem1* knockdown (KD) ESC lines. We kindly refer the Reviewer to this publication for a thorough validation of the KD lines; nonetheless, we provide a point-by-point response to the specific comments below:

No validation of the KD is provided.

The generation of the *T-UCstem1* KD ESC clones used in this study was previously reported in Fiorenzano et al. (Fiorenzano A, et al. Stem Cell Reports. 2018 Mar 13;10(3):1102-1114. doi: 10.1016/j.stemcr.2018.01.014), where the clones were thoroughly characterized and the knockdown was validated.

Clones 1 and 2 have slightly different phenotypes.

The observed phenotypic differences between KD-1 and KD-2 clones are due to subtle variations in KD efficiency, which is consistent with that reported in Fiorenzano et al., 2018 (Fiorenzano A, et al. Stem Cell Reports. 2018 Mar 13;10(3):1102-1114. doi: 10.1016/j.stemcr.2018.01.014). These clones were selected to reflect different levels of *T-UCstem1* knockdown, and this variability is critical for understanding the dose-dependent effects of *T-UCstem1* on ESC behavior.

How long does the KD last?

Both KD-1 and KD-2 are stable cell lines generated using shRNA-mediated knockdown. The details are reported in Fiorenzano et al., 2018.

The starting KD ESCs are also not characterized (beyond a colony formation assay). Are there gene expression changes already in the naïve state or do these only manifest upon naïve pluripotency exit?

The molecular and functional characterization of the KD ESCs, including pluripotency state, proliferation rate and gene expression changes, is reported in Fiorenzano et al., 2018 (Fiorenzano A, et al. Stem Cell Reports. 2018 Mar 13;10(3):1102-1114. doi: 10.1016/j.stemcr.2018.01.014).

Briefly, under standard FBS/LIF conditions in the presence of feeders, *T-UCstem1* KD ESCs generate flat colonies and exhibit enhanced neuronal differentiation.

EMBOJ-2024-119368-T

Conversely, the addition of *2i* to the culture medium is sufficient to rescue the dome-shaped colony morphology of KD cells, which also stained positive for alkaline phosphatase (see Figure 4E of Fiorenzano et al.). This morphological rescue was further supported by molecular data (western blotting) indicating restored expression of key pluripotency markers such as Nanog and Sox2 (Figure 4F in Fiorenzano et al., 2018).

Throughout the study, the cells are cultured at low density in FBS/LIF/*2i* prior to the gastruloid formation assay, a condition under which KD cells show no differences in pluripotency state compared to the control.

What is the rate of proliferation of control and KD cells in naïve conditions and gastruloids?

Under FBS/LIF/*2i* culture conditions KD and control cells do not show differences in proliferation rate. Furthermore, the diameter of NT and KD aggregates at 48 hours is comparable (Figure 1C and D), thus further ruling out significant differences in proliferation.

Do T-UCstem1 levels increase by 96-120 hours of the gastruloid protocol?

We have analyzed the expression of *T-UCstem1* during gastruloid formation by Q-RT PCR (see revised manuscript, page 13, lines 292-294, and Figure EV6A). These data indicate that *T-UCstem1* is expressed during gastruloid development with expression progressively decreasing between 48 and 120 hrs. Importantly, the expression profile of *T-UCstem1* shows an inverse correlation with that of DKK-1, which conversely increases over the same time period, supporting our hypothesis that *T-UCstem1* negatively regulates DKK-1 and subsequently modulates WNT signaling.

The authors would also need to rescue the phenotype by re-introducing T-UCstem1.

To address the Reviewer's comment, we performed a rescue experiment by overexpressing the ultra-conserved element uc.170+ in both *T-UCstem1* knockdown ESC clones (KD1 and KD2). Specifically, we used a bicistronic expression plasmid encoding both uc.170+ and GFP under the control of the ubiquitous CAG promoter that has been previously described (Pascale E, et al. *Stem Cell Reports*. 2020;15(4):836–844. doi:10.1016/j.stemcr.2020.08.009). A plasmid expressing only GFP served as a control. Using these constructs, we generated four new ESC lines: KD-1^{GFP}, KD-1^{T-UC}, KD-2^{GFP}, and KD-2^{T-UC}.

Notably, re-expression of the lncRNA in both KD1 and KD2 ESC lines restored DKK-1 expression to wild-type levels and fully rescued the ability of KD1 and KD2 ESCs to generate elongated gastruloids in the presence of 3 μ M CHIR. This rescue experiment confirms that the phenotype observed in the KD lines is specifically due to the knockdown of *T-UCstem1*. These Results are now shown in Figure 9 and reported on page 13, lines 294-298 of the revised manuscript.

EMBOJ-2024-119368-T

We are grateful to the Reviewer for suggesting these important analyses, which have further strengthened the conclusions of our study.

2. Reproducibility: throughout the article, the number of gastruloids analyzed and the number of independent experiments are not mentioned.

We thank the reviewer for pointing out some weakness and highlighting the missing important information. Following the Reviewer's comment, we have carefully reviewed our manuscript and realized that, although the number of gastruloids analyzed and the number of independent experiments were included in most figure legends, this information was occasionally missing. We have now carefully revised all the figure legends to consistently include the exact number of gastruloids analyzed and the number of replicates performed.

The IF images are not analyzed. Therefore, the reproducibility of the images shown and/or variability of the phenotype is not clear.

The representative images shown were selected from multiple independent experiments (with an n ranging from 10 to 20 gastruloids per condition, as reported in each figure legend in the revised manuscript) and consistently reflect the observed phenotype, with no major variability across replicates. We did not quantify signal intensity, as our primary aim was to assess the spatial distribution of marker expression rather than its absolute levels.

In the CHIR rescue only OCT4 and CDX2 expression are shown.

To strengthen the CHIR rescue experiment, we extended our analysis to include additional pluripotency and differentiation markers. Specifically, we now include immunofluorescence analysis for Nanog and Brachyury (see revised manuscript, page 8, line 169, and Figure 4B).

A complete analysis of markers and single cell quantification would be needed to support the rescue experiment

While we fully acknowledge the value of such an approach, performing single-cell analysis would require considerable time and financial resources, and in our view, it would not significantly enhance or alter the conclusions drawn from our current data. Instead, we prioritized functional experiments and targeted analyses that directly address our main research questions, as detailed in the manuscript.

Moreover, the methods section lacks enough detail (e.g., media recipes, catalog numbers, etc).

In response to the Reviewer's comment, we have thoroughly revised and expanded the Methods section. In addition, we have completed the '*Reagents Tools Table*' in accordance with EMBO Journal guidelines.

3. The model the authors propose is that increased expression of DKK1 in T-UCstem1 KD cells is responsible for the defects observed in the gastruloids.

EMBOJ-2024-119368-T

What are the levels of Wnt activity? Using a Wnt reporter ESC line would be informative. This is important as Brachyury is expressed in the KD EBs, which suggests Wnt signaling is active.

We agree with the Reviewer that WNT signaling is active in KD cells, and we do not suggest otherwise. Our model does not suggest a complete loss of WNT activity, but rather an attenuation of its signaling levels mediated by the upregulation of the WNT inhibitor DKK-1. This eventually results in the need of higher concentration of CHIR to achieve a comparable cellular response, especially in terms of elongation and patterning. This notion is supported by rescue experiments, in which KD-2—exhibiting a greater upregulation of DKK-1—requires a higher CHIR concentration (6 μ M) to effectively restore the phenotype.

Moreover, the increase in DKK1 expression shown in Figure 7D seems to be small in clone 1.

Indeed, the lower increase in DKK-1 expression in KD-1 compared to KD-2, aligns with the differences in knockdown efficiency between the two clones, with KD-1 exhibiting a lower knockdown efficiency than KD-2 (see Fiorenzano et al., 2018). Nevertheless, both clones consistently show phenotypic defects that are responsive to WNT modulation, reinforcing the central role of DKK1-mediated WNT inhibition in the observed phenotype.

Together, these findings support the model by which *T-UCstem1* regulates the fine-tuning of WNT signaling through DKK-1, rather than acting as an on/off switch for pathway activity.

What are the protein levels of DKK1?

To address the Reviewer's comment, we attempted to measure DKK-1 protein levels by Western blot using several commercially available DKK-1 antibodies. Unfortunately, all of our efforts were unsuccessful, likely due to technical issues. Since DKK-1 is a secreted protein, we also tested conditioned medium from both undifferentiated ESCs and gastruloids; however, we were unable to obtain reliable results, suggesting that DKK-1 protein most likely fall below the detection threshold under our experimental conditions.

Despite this, we believe that the increased expression of DKK-1 in both KD clones, as measured by qRT-PCR and recovered upon uc170 overexpression in the rescue experiments, along with the functional assays—including the chimeric gastruloid assay and pharmacological inhibition of DKK-1—strongly support our conclusion.

The DKK1 experiments (Figure 8) should also show a detailed molecular characterization.

To address the Reviewer's comment, we have included a comprehensive new set of immunofluorescence analyses for the DKK-1 experiments in the revised Figure 8. Specifically, we analyze: (i) for the experiment of addition of recombinant DKK-1 protein to Control ESC aggregates, the expression of Nanog and Cdx2 (Figure

EMBOJ-2024-119368-T

EV5A); (ii) for the experiments of treatment of KD aggregates with WAY262611, the expression of Oct4, Nanog, Brachyury and Cdx2 (Figure 8G and EV5B).

4. Defects in pluripotency exit: from the data presented is not clear whether loss of *T-UCstem1* causes a naïve pluripotency exit defect or a defect in lineage specification. The colony formation assay in 2iLIF tests for naïve pluripotency, while the IF data presented by the authors analyses Oct4, which is a core pluripotency marker. What is exactly happening in terms of pluripotency progression? Is naïve pluripotency exit compromised? Are there cells expressing naïve pluripotency markers in KD gastruloids? Are formative pluripotency markers expressed? The pluripotent status of the cells should be clearly defined.

Following the Reviewer's comment, we have clarified this important issue.

Different lines of evidence suggest that the loss of *T-UCstem1* impairs exit from naïve pluripotency. It is important to consider, however, that disentangling naïve pluripotency exit from lineage specification is inherently challenging, as the two processes are tightly interconnected and often occur in a sequential and partially overlapping manner. First, as also indicated by the Reviewer, the colony formation assay in FBS/LIF/2i (Figure 6B), which tests for naïve pluripotency, shows a marked increase in the fraction of domed-shaped and AP positive colonies in the cell population derived from *T-UCstem1* KD aggregates compared to control. This observation is further supported the single-cell RNA sequencing data, which reveals an enrichment of a population that in the original manuscript was annotated as *pluripotent* in the *T-UCstem1* KD condition.

However, we recognized that labeling this population as "pluripotent" may lead to confusion. These cells were annotated based on Suppinger et al., 2023 (Suppinger et al., 2023, Cell Stem Cell 30, 867–884. doi.org/10.1016/j.stem.2023.04.018), and correspond to a naïve pluripotent state (details are reported in the Methods section page 23, lines 514-520). To clarify this point, we now explicitly label this population as "naïve pluripotent" (see revised manuscript, page 9-10, lines 205-226, and Figures 5D-F and EV4C,D). We also examined the expression of formative pluripotency markers, but did not observe any significant enrichment.

Finally, it is important to note that while most cells fall in the naïve pluripotent population, a subset of them appears to retain the naïve pluripotent signature while also initiating lineage-specific programs. This results in the emergence of an aberrant cell population co-expressing naïve pluripotency markers and early lineage commitment markers, indicative of an incomplete and dysregulated exit from pluripotency.

EMBOJ-2024-119368-T

5. Mechanism: how does T-UCstem1 control DKK1 expression? Some level of mechanistic insight should be expected for publication in the EMBO journal.

We have previously reported that *T-UCstem1* interacts with and stabilizes the PRC2 complex on bivalent chromatin domains to modulate gene expression in undifferentiated ESCs (Fiorenzano et al; SCR 2018). Thus, *T-UCstem1* may regulate DKK-1 expression through a similar PRC2-dependent mechanism. However, we cannot rule out that additional regulatory mechanisms may be involved, including the possibility of a direct interaction between *T-UCstem1* and the DKK-1 locus or transcript. Further in-depth investigation and studies are needed to explore these possibilities, which will undoubtedly be of significant relevance.

Minor comments:

1. The manuscript needs careful editing.

We have conducted a thorough rereading and editing to improve clarity and coherence.

2. The limitations of gastruloids should also be acknowledged in the introduction.

We have acknowledged the limitations of gastruloids in the Introduction of the revised manuscript (page 3, lines 54-57).

3. Figure 5F: similar plots should be shown for the control gastruloids.

We thank the reviewer for this suggestion. However, the plots in Figure 5F were specifically designed to analyze the naïve pluripotent cell population that emerges uniquely in the *T-UCstem1* KD gastruloids. Since this population is absent in the non-targeting (NT) control gastruloids, we did not generate equivalent plots for the control condition.

4. Figure 6D: the use of the cytospin is not ideal. If possible, the authors should image and quantify specific planes within the gastruloids. The study would also benefit from higher magnification and resolution images.

We chose to perform the double staining analysis shown in Figure 6D on dissociated gastruloids in order to allow for a more precise and unambiguous assessment of Oct4/Brachyury co-expression at the single-cell level. While wholemound imaging provides valuable spatial context, it can sometimes make it difficult to definitively determine whether two markers are co-expressed within the same cell due to overlapping signals in densely packed regions. By dissociating the gastruloids, we ensured a clearer resolution and more confident identification of double-positive cells. This approach complemented the wholemound analyses presented elsewhere in the manuscript and strengthened our interpretation of co-expression dynamics.

EMBOJ-2024-119368-T

5. The authors conclude that the KD gastruloids display a wide range of shapes and sizes but this is true only for clone 2, not clone 1.

The wide range of shapes and sizes is appreciable from the images shown in Figure EV2B.

6. Line 446: The elevated levels of DKK1 found in patients with unexplained miscarriages are seen in the endometrium, not the embryo. This statement is misleading.

We thank the Reviewer for this important clarification. We agree that elevated DKK-1 levels in patients with unexplained miscarriages have been reported in the endometrium, not in the embryo itself. However, considering that DKK-1 is a secreted, soluble factor, we believe that its increased presence in the endometrium may still have a paracrine effect on the embryo. Therefore, we consider this observation to be in line with the central findings of our work.

Dear Annalisa,

Thank you again for submitting your revised manuscript (EMBOJ-2024-119368R) to The EMBO Journal for our consideration, and for your patience during peer review. As I have already informed you, your manuscript has been sent back for re-review to the original referees #1 and #3, who had previously assessed the initial version of your work, and we have received their comments that I have already shared with you (included again below). I would also like to thank you for the point-by-point response to these comments that you shared with me, which was very helpful for us to reach a well-informed and balanced decision on your manuscript.

I am pleased to say that both referees find the revised manuscript considerably improved and mention that the majority of the initially raised criticisms and concerns have been successfully addressed. However, they both point out that a number of remaining issues require correction or clarification before the manuscript can be accepted for publication. In particular, they raise concerns regarding statistical analysis and presentation of the results, data reproducibility, and annotation and validation of scRNA-seq data.

In light of these comments, and also considering your responses and described corrections included in your point-by-point response, I would like to invite you to submit a final version of your manuscript fully addressing all remaining comments of the two referees, along the lines of your point-by-point response and considering our feedback as follows:

Ref. #1, point 1: please make sure that the raw deltaCT values for the qRT-PCR experiments are also included in the uploaded Source Data for the respective Figures. Please also make sure that biological (not technical) replicates are considered in statistical comparisons, and that these are performed using the correct statistical tests, to avoid pseudoreplication and the risk of type I errors (see also below).

Ref. #1, point 2: please revise the manuscript making the necessary corrections according to your point-by-point response.

Ref. #3, point 1: please follow the referee's suggestion to specify that you use the exact same line, and that the KD efficiency was characterized in the previous, cited work. Please also add your comments on the different KD levels.

Ref. #3, point 2: the number of gastruloids imaged in each immunofluorescence experiment should be provided in the respective Figure legends (see also below). If there were any gastruloids showing a different marker localization, their number should also be provided.

Ref. #3, point 3: please make the necessary corrections according to your point-by-point response (see also below).

Ref. #3, point 4: we concur with the referee that:

- i. when there are more than 2 conditions, pairwise comparisons using Student's t-tests is not a valid approach, since it could increase the risk of false positives (type I errors); other methods suitable for multiple comparisons, followed by appropriate post-hoc tests, might have to be used instead, and
- ii. no statistics can be calculated and shown when $n=2$ (biological replicates); including the technical replicates to increase the sample size is an example of pseudoreplication and must be avoided, as it artificially inflates the sample size leading to unreliable statistical comparisons; for such comparisons to be valid, more biological replicates must be analyzed; for experiments where this is not necessary, the individual values of the two biological replicates should be shown in all Figures, and no statistical comparisons can be performed and shown/discussed.

Please include in your resubmission a detailed point-by-point response fully addressing all remaining referees' comments and describing any changes to the manuscript.

From the editorial side, there are also a few changes and corrections we need you to make in the manuscript before we can proceed with its handling further:

- Please mark the corresponding authors on the title page of your revised manuscript using asterisks or another symbol.
- Thank you for providing access information to the bulk RNA sequencing data generated in your study. We were not able to locate the single-cell RNA sequencing data, however, and we kindly ask you to make sure that they are also deposited to an appropriate repository and their access information is provided in the Data availability section of your revised manuscript.
- A conflict-of-interest statement, with heading "Disclosure and competing interests statement", and placed below the Acknowledgements section, is mandatory for all research articles. Please find more information in our guide to authors: <https://www.embopress.org/page/journal/14602075/authorguide#conflictsofinterest>.
- The author contributions statement should be removed from the manuscript file. Instead, we use CRediT to specify the contributions of each author in the journal submission system. Please feel free to use the free text box to provide more detailed descriptions during submission. See also our guide to authors for more information: <https://www.embopress.org/page/journal/14602075/authorguide#authorshippinguidelines>.
- Please note that DOI identifiers should only be provided in the list of citations for preprints and datasets that have not been

published yet; the section heading needs renaming to "References".

- Please make sure that section "Experimental study design and statistics" in the Author Checklist is completed as appropriate.

- We noticed that callouts for Figure 9D are missing; "supplementary" (page 23) should not be used; callouts are also missing for the Appendix Figures; Tables 1 and 2 are called out but seem to be missing.

- The Excel file "Appendix Table Annotation Scores" should be renamed to "Dataset EV1" and called out where appropriate in the manuscript. Its legend should be provided in a separate tab/sheet in the same Excel file.

- The Appendix PDF file should have on its first (title) page the heading "Appendix for" followed by the manuscript's title and a brief Table of Contents including page numbers for all listed items. The nomenclature needs correction throughout the file to "Appendix Figure S1-S4" and the Figures included in the Appendix need to be called out in the main manuscript file.

- Please note that EMBO press papers are accompanied online by:

A) a short (2 sentences) summary of the findings and their significance,

B) 2-5 short bullet points highlighting the key results, and

C) a synopsis image in .jpg or .png format that is exactly 550 pixels wide and 300-600 pixels high (the height is variable). Please note that all text needs to be legible at the final size.

Please upload this information along with your revised manuscript (the text for A and B should be provided in a separate Word file).

- During our standard pre-acceptance Figure checks, we detected possible cell reuse:

1. Between Figure 1F & Figure 3C (not listed in the figure legend).

2. Between Figure 1F & Figure 8G (not listed in the figure legend).

3. Within Figure 2C.

4. Between Figure 4A & Figure EV3C (not listed in the figure legend).

5. Between Figure 7 B&C & Figure EV1B (not listed in the figure legend).

6. Between Figure 8F & Figure EV1B (not listed in the figure legend).

7. Between Figure 9B & Figure EV1B (not listed in the figure legend).

8. Within Figure EV1B.

9. Between Figure EV1B & Figure EV3 A&C (not listed in the figure legend).

10. Between Figure EV3D & Appendix Figure S3D (not listed in the figure legend).

Please check these Figures again and confirm all reuse. Figures need to be corrected or the reuse needs to be detailed in the corresponding Figure legends if it is intentional and justified.

- During our routine data checks, our data editors have raised the following queries regarding figures, data, and legends. Please make sure that all requests below are completely addressed in the final version of your manuscript (please highlight all changes in the manuscript):

1. Please indicate the statistical test used for data analysis in the legends of Figures 1C, E; 2B, 3A, B; 4A, 5B, 6B, C; 7B-D; 8D-F; 9C, EV2 A-D; EV4 A, EV5B, C; EV6 A, B; S2A-D, S4A.

2. Please note that the box plots need to be defined in terms of minima, maxima, centre, bounds of box and whiskers, and percentile in the legends of Figures 1C, E; 2B, 6B, C; 7B, C; 8D-F; 9C, EV2 A-D; EV5 B, C; S2 A-D.

3. Please note that information related to "n" is missing in the legends of Figures 7E, EV4 A, S2 A-D; S4A.

Please also note that as part of the EMBO publications' Transparent Editorial Process, The EMBO Journal publishes online a Peer Review File along with each accepted manuscript. This File will be published in conjunction with your paper and will include the referee reports, your point-by-point response and all pertinent correspondence relating to the manuscript. You can opt out of this by letting the editorial office know (contact@embojournal.org). If you do opt out, the Peer Review File link will point to the following statement: "No Peer Review File is available with this article, as the authors have chosen not to make the review process public in this case."

We look forward to seeing a final version of your manuscript as soon as possible. Please let us know if you have any questions and use this link to submit your revision: <https://emboj.msubmit.net/cgi-bin/main.plex>.

Best regards,

Ioannis

Ioannis Papaioannou, PhD
Editor, The EMBO Journal

Referee #1:

The authors have made commendable progress in addressing many of the previous concerns. In particular, documenting the expression of T-UCSTEM and DKK1 during gastruloid development, along with the rescue experiment in both knockdown lines, strongly supports the functional relevance of T-UCSTEM in the observed phenotype. These additions significantly strengthen the manuscript. However, there are still key points that require further clarification and substantiation before the study can be considered ready for publication.

1. The qRT-PCR (Fig EV6A-B) data need to be presented with greater clarity and transparency.

In Fig. EV6A, the reported standard deviations are extremely low, which is unexpected given the typical variability in gastruloid-based experiments. This raises concerns about whether the plots reflect technical or biological replicates, and how variability was quantified. Even technical replicates often show more dispersion than is visible here.

Some aspects of the statistical analysis are difficult to reconcile. For example, a ~10-fold increase in expression between 0h and 120h yields a p-value of 0.062, while a smaller change (~5-fold at 48h) gives a p-value of 0.018, despite similarly low variation. In Fig. EV6B, small differences between KD1 and KD1-GFP yield quite different p-values, and KD2 and KD2-GFP, despite visibly different values, have identical p-values.

These inconsistencies suggest either an issue with how variation is represented or with the statistical comparisons themselves. After double checking this analysis, the authors should include raw CT values and describe in detail how normalization, statistical tests, and replicate definitions were handled. This is essential for evaluating the robustness of the results.

2. Single-cell RNA-seq Annotation and Validation

The choice to use an annotation transfer approach from reference atlases (Pijuan-Sala and Suppinger) is a reasonable one in principle. However, the current implementation lacks the necessary validation and documentation to support the conclusions drawn from the data.

There is no clear assessment of how well the transferred annotations match expected biology. Without validation using known marker genes, it is not possible to evaluate the accuracy of the cluster identities. This is a critical omission, especially given that the results include non-intuitive and unexpected cell identities based on the known gastruloid cell composition.

Furthermore, the single-cell data are not currently available in the GEO repository, and the code used for the analysis has not been shared. Both are required for reproducibility and transparency.

At a bare minimum, the authors should provide a supplementary figure showing the expression of canonical markers used to define the transferred identities. This would allow readers to assess whether the annotations are biologically meaningful and consistent with the atlases.

If the transferred annotations do not correspond well to marker gene expression, the authors will need to comprehensively revisit the analysis. This includes careful QC of the reference atlases and the dataset, validation of cluster identities, and potentially re-annotation using marker-based or semi-supervised approaches.

As it stands, the single-cell analysis is the least developed part of the manuscript. If the authors are unable to provide the necessary validation, I would recommend either removing this section or clearly limiting its interpretation. I would still recommend publication in absence of this dataset. But I do not believe this analysis should remain present in its current form.

In summary, the manuscript presents strong experimental evidence for the role of T-UCSTEM in gastruloid development. The rescue experiment is particularly compelling. However, the conclusions drawn from the qPCR and single-cell data require further clarification, transparency, and validation. Once these issues are addressed, particularly through the provision of raw data and annotation validation, I would be happy to support publication.

Referee #3:

In this revised version of the manuscript, Coppola et al have addressed some of the reviewers' comments. I think there was a

clear consensus among the reviewers of the need to rescue the phenotype and characterise the expression pattern of T-UCstem1. This has now been done. However, there are still important points that need to be clarified, especially in terms of the reproducibility of the data:

1. T-UCstem1 KD: The manuscript heavily relies on the data from Fiorenzano et al, 2018. The authors mention that the KD lines have already been described in this manuscript. Still, they should clearly specify that they are using the exact same lines, and that the KD efficiency was characterised in that previous work. They should also comment on the different KD levels that are reached in the two mESC lines.

2. Analysis of the IF data: I was not expecting a quantification of the IF intensity, as, of course, the pattern of expression is the key parameter. What the authors should provide for every single IF panel (and they have not) is the number of gastruloids per condition that show a polarised localisation of the marker that is being analysed out of the total number of gastruloids that have been imaged.

3. N numbers: The authors now provide n numbers in the figure legends, but for several experiments, the n numbers mentioned in the legends do not match the n numbers shown in the plots. For example, in Figure 4A, the legend mentions 20 gastruloids per condition, but the graph does not show 20 data points for any of the conditions. Another example, the legend of Figure 9C specifies 14 gastruloids per condition, but the graph shows many more data points.

4. Statistics: The methods section mentions that a Student's t-test has been performed to analyse the data. This is incorrect for all the experiments, as in all cases, there are more than 2 conditions. Moreover, in Figure EV6, only 2 samples per condition have been analysed. How have the authors performed a statistical analysis with just $n=2$? Additional samples are needed to properly complete these experiments.

EMBOJ-2024-119368R

Reviewer reports:

Referee #1: The authors have made commendable progress in addressing many of the previous concerns. In particular, documenting the expression of T-UCSTEM and DKK1 during gastruloid development, along with the rescue experiment in both knockdown lines, strongly supports the functional relevance of T-UCSTEM in the observed phenotype. These additions significantly strengthen the manuscript. However, there are still key points that require further clarification and substantiation before the study can be considered ready for publication.

We thank the reviewer for the positive comments and for raising constructive criticisms, which we have carefully considered to improve the manuscript. We have addressed the specific issues raised, as detailed below

1. The qRT-PCR (Fig EV6A-B) data need to be presented with greater clarity and transparency.

In Fig. EV6A, the reported standard deviations are extremely low, which is unexpected given the typical variability in gastruloid-based experiments. This raises concerns about whether the plots reflect technical or biological replicates, and how variability was quantified. Even technical replicates often show more dispersion than is visible here.

Some aspects of the statistical analysis are difficult to reconcile. For example, a ~10-fold increase in expression between 0h and 120h yields a p-value of 0.062, while a smaller change (~5-fold at 48h) gives a p-value of 0.018, despite similarly low variation. In Fig. EV6B, small differences between KD1 and KD1-GFP yield quite different p-values, and KD2 and KD2-GFP, despite visibly different values, have identical p-values.

These inconsistencies suggest either an issue with how variation is represented or with the statistical comparisons themselves. After double checking this analysis, the authors should include raw CT values and describe in detail how normalization, statistical tests, and replicate definitions were handled. This is essential for evaluating the robustness of the results.

We are very grateful to the reviewer for bringing this inconsistency to our attention. After carefully double-checking our analysis, we identified an error in the qRT-PCR data within the Excel file; specifically, a mistake in the calculation of the standard deviation. We sincerely apologize and we thank the reviewer for giving us the opportunity to correct the error.

To address the reviewer's comment, we have reanalysed the qRT-PCR data and revised Figure EV6A and EV6B, accordingly.

We would like to clarify that the data shown in Fig. EV6A-B are based on biological replicates. For each biological replicate, we performed technical triplicates. We first calculated the average of the technical replicates for each gene (both the gene of interest and the housekeeping gene). We then computed ΔCT values, followed by the average ΔCT across biological replicates. Next, we apply the formula $2^{(-\Delta CT * 1000)}$, that are the values reported in the graph. Statistical comparisons were performed on these averaged biological replicate values.

EMBOJ-2024-119368R

The raw Δ CT values were not initially included, as they were placed in a supplementary file. In response to the reviewer's request, we have now added all raw Δ CT values to the Source Data.

Furthermore, in response to Reviewer #3's comment and to improve the robustness of our statistical analysis, we have reanalyzed all the data using one-way ANOVA followed by Tukey's multiple comparison test. Accordingly, in the revised Figure 7D, the qRT-PCR data have been updated to include an additional biological replicate (n=3), and the statistical analysis was performed using one-way ANOVA followed by Tukey's multiple comparison test.

2. Single-cell RNA-seq Annotation and Validation

The choice to use an annotation transfer approach from reference atlases (Pijuan-Sala and Suppinger) is a reasonable one in principle. However, the current implementation lacks the necessary validation and documentation to support the conclusions drawn from the data.

There is no clear assessment of how well the transferred annotations match expected biology. Without validation using known marker genes, it is not possible to evaluate the accuracy of the cluster identities. This is a critical omission, especially given that the results include non-intuitive and unexpected cell identities based on the known gastruloid cell composition.

Furthermore, the single-cell data are not currently available in the GEO repository, and the code used for the analysis has not been shared. Both are required for reproducibility and transparency.

At a bare minimum, the authors should provide a supplementary figure showing the expression of canonical markers used to define the transferred identities. This would allow readers to assess whether the annotations are biologically meaningful and consistent with the atlases.

If the transferred annotations do not correspond well to marker gene expression, the authors will need to comprehensively revisit the analysis. This includes careful QC of the reference atlases and the dataset, validation of cluster identities, and potentially re-annotation using marker-based or semi-supervised approaches.

As it stands, the single-cell analysis is the least developed part of the manuscript. If the authors are unable to provide the necessary validation, I would recommend either removing this section or clearly limiting its interpretation. I would still recommend publication in absence of this dataset. But I do not believe this analysis should remain present in its current form.

We thank the reviewer for raising these points and for the valuable suggestions regarding the single-cell analysis.

First, we sincerely apologize for the oversight regarding data availability. While the ArrayExpress E-MTAB-14503 token provided at first submission remains unchanged, we inadvertently omitted it from the revised manuscript's Data Availability section. We have

EMBOJ-2024-119368R

corrected this issue and ensured that the ArrayExpress accession number is clearly stated in the revised manuscript.

Regarding annotation, we have taken on board the reviewer's concerns about the non-intuitive and unexpected cell identities. We have limited the annotation to only the lists derived from gastruloids according to Suppinger et al. This approach repaid in terms of simplicity and possibly accuracy of the annotation. Importantly, none of the 'non-intuitive and unexpected' populations show significant changes in the *T-UCstem1* knockdown condition and thus do not affect our core findings regarding the role of this lncRNA. Anyway, thanks to the reviewer, we are now able to label those problematic clusters simply with labels coherent with gastruloid literature (Suppinger et al.).

These results are now shown in the revised Figure 5 and described in the revised version of the manuscript (pages 9 and 10).

As by reviewer request, here we pinpoint the most important passages of the single cell analysis:

1. We created a `seuratObjList` with a `seuratObj NT` and a `seuratObj KD1` (`cells=3,min.features=200`).
2. We kept the cells with less than 15% of mitochondrial genes while cells with few or too many genes were removed.
3. We merged the two samples, normalized the data, found the variable features (genes), scaled the data and make the PCA.
4. We integrated layers, according to Seurat v5 pipeline, using the RPCA method and then jointed the layers.
5. Finally, we looked for neighbours (construct a KNN graph based on the euclidean distance in PCA space), found clusters (apply modularity optimization techniques such as the Louvain algorithm) and run UMAP (a non-linear dimensional reduction techniques), all using default parameters.
6. Annotation made according to ScType algorithm (Ianevski et al., 2022) given a list of 19 different gastruloids cell population markers from Suppinger et al., 2023.
7. After the down sampling of the NT gastruloids (to easy the visual comparison with KD1), the uniform manifold approximation and projection (UMAP) algorithm used to depict the cell populations and the expression of specific makers together with violin plot to show DKK1 expression.
8. For each cell population Fisher test performed to pinpoint the significant odd ratios (OR) (FDR-adjusted p-value <0.01) in KD-1 vs NT gastruloids.

The same information is reported in the Method section (page 23 lines 517-525).

In summary, the manuscript presents strong experimental evidence for the role of T-UCSTEM in gastruloid development. The rescue experiment is particularly compelling. However, the conclusions drawn from the qPCR and single-cell data require further clarification, transparency, and validation. Once these issues are addressed, particularly through the provision of raw data and annotation validation, I would be happy to support publication.

EMBOJ-2024-119368R

Referee #3: In this revised version of the manuscript, Coppola et al have addressed some of the reviewers' comments. I think there was a clear consensus among the reviewers of the need to rescue the phenotype and characterise the expression pattern of T-UCstem1. This has now been done. However, there are still important points that need to be clarified, especially in terms of the reproducibility of the data:

1. T-UCstem1 KD: The manuscript heavily relies on the data from Fiorenzano et al, 2018. The authors mention that the KD lines have already been described in this manuscript. Still, they should clearly specify that they are using the exact same lines, and that the KD efficiency was characterised in that previous work. They should also comment on the different KD levels that are reached in the two mESC lines.

Following the reviewer's comment, we have now specified in the Methods section (page 18 lines 390-396) that the exact same T-UCstem1 knockdown (KD) ESC lines described in Fiorenzano et al., 2018, were used in the present study. We also clarified that the KD efficiency was thoroughly characterized in that previous study. Additionally, we have emphasized that the two KD clones (KD-1 and KD-2) were selected based on their distinct levels of T-UCstem1 silencing, allowing us to better explore potential dose-dependent effects on ESC behavior.

2. Analysis of the IF data: I was not expecting a quantification of the IF intensity, as, of course, the pattern of expression is the key parameter. What the authors should provide for every single IF panel (and they have not) is the number of gastruloids per condition that show a polarised localisation of the marker that is being analysed out of the total number of gastruloids that have been imaged.

We have updated the figure legends in the revised version of the manuscript to include, for each immunofluorescence panel, the number of gastruloids per condition that were imaged.

3. N numbers: The authors now provide n numbers in the figure legends, but for several experiments, the n numbers mentioned in the legends do not match the n numbers shown in the plots. For example, in Figure 4A, the legend mentions 20 gastruloids per condition, but the graph does not show 20 data points for any of the conditions. Another example, the legend of Figure 9C specifies 14 gastruloids per condition, but the graph shows many more data points.

We thank the reviewer for raising this important point. We have carefully double-checked the sample sizes (N numbers) across all figures and figure legends, and we agree with the reviewer's observations: 1) In Figure 4A, 10 gastruloids per condition were analysed and reported in the graph. We have now corrected the corresponding figure legend to reflect this accurately. 2) In Figure 9C, 14 gastruloids per condition were analysed. This has been corrected and included in the revised Figure.

4. Statistics: The methods section mentions that a Student's t-test has been performed to analyse the data. This is incorrect for all the experiments, as in all cases, there are more than 2 conditions. Moreover, in Figure EV6, only 2 samples per condition have been analysed. How have the authors performed a statistical analysis with just n=2? Additional samples are needed to properly complete these experiments.

EMBOJ-2024-119368R

We have revised all statistical analyses using one-way ANOVA with Tukey's multiple comparison test, as now reported in the Methods section (page 24, lines 532–540).
Regarding

Figure EV6A, only two biological replicates were available; therefore, we have reported the individual values of these replicates without performing statistical comparisons. We did not perform a third biological replicate considering the consistent outcomes of the duplicates and due to practical constraints, since RNA preparation from gastruloids requires a large number of samples, especially at early time points such as 48 and 72 hours, making additional replicates time- and resource-intensive. Conversely, the qRT-PCR data shown in Figure EV6B have been updated to include an additional biological replicate (n=3), and the statistical analysis was performed using one-way ANOVA followed by Tukey's multiple comparison test.

All editorial and formatting issues were resolved by the authors.

Dear Annalisa,

Congratulations on an excellent manuscript! I am very pleased to inform you that we have now received the comments of referee #1 (included below), who is satisfied with how all remaining concerns have now been addressed, and your manuscript has therefore been accepted for publication in The EMBO Journal. Thank you for comprehensively addressing all initially raised concerns and our editorial requests for corrections and changes.

If you have any questions, please do not hesitate to contact the Editorial Office. Thank you for your contribution to The EMBO Journal. Working with you has been a pleasure!

Best wishes,

Ioannis

Referee #1:

I think the manuscript is now ready for publication. I commend the authors for their efforts in ensuring the solidity of the conclusions from their paper.
